# Tumour mutations in long noncoding RNAs enhance cell fitness

Roberta Esposito [1,2,3,16] ✉, Andrés Lanzós[1,2,4,16], Tina Uroda[5,6], Sunandini Ramnarayanan[5,6,7], Isabel Büchi[2,8], Taisia Polidori[1,2], Hugo Guillen-Ramirez[5,6], Ante Mihaljevic[1,2], Bernard Mefi Merlin[1,2], Lia Mela[1,2], Eugenio Zoni[2,9], Lusine Hovhannisyan [2,10], Finn McCluggage [11,12], Matúš Medo [2,10], Giulia Basile[1,2], Dominik F. Meise[1,2], Sandra Zwyssig [1,2], Corina Wenger[1,2], Kyriakos Schwarz[1,2], Adrienne Vancura[1,2], Núria Bosch-Guiteras[1,2,4], Álvaro Andrades [13,14,15], Ai Ming Tham[5,6], Michaela Roemmele[1,2], Pedro P. Medina[13,14,15], Adrian F. Ochsenbein [1,2], Carsten Riether[1,2], Marianna Kruithof-de Julio [2,9], Yitzhak Zimmer[2,10], Michaela Medová[2,10], Deborah Stroka [2,8], Archa Fox[11,12] & Rory Johnson [1,2,5,6] ✉

Long noncoding RNAs (lncRNAs) are linked to cancer via pathogenic changes in their expression levels. Yet, it remains unclear whether lncRNAs can also impact tumour cell fitness via function-altering somatic "driver" mutations. To search for such driver-lncRNAs, we here perform a genome-wide analysis of fitness-altering single nucleotide variants (SNVs) across a cohort of 2583 primary and 3527 metastatic tumours. The resulting 54 mutated and positively-selected lncRNAs are significantly enriched for previously-reported cancer genes and a range of clinical and genomic features. A number of these lncRNAs promote tumour cell proliferation when overexpressed in in vitro models. Our results also highlight a dense SNV hotspot in the widely-studied *NEAT1* oncogene. To directly evaluate the functional significance of *NEAT1* SNVs, we use in cellulo mutagenesis to introduce tumour-like mutations in the gene and observe a significant and reproducible increase in cell fitness, both in vitro and in a mouse model. Mechanistic studies reveal that SNVs remodel the *NEAT1* ribonucleoprotein and boost subnuclear paraspeckles. In summary, this work demonstrates the utility of driver analysis for mapping cancer-promoting lncRNAs, and provides experimental evidence that somatic mutations can act through lncRNAs to enhance pathological cancer cell fitness.

Tumours arise and develop via somatic mutations that confer a fitness advantage on cells[1]. Such driver mutations exert their phenotypic effect by altering the function of genes or genomic elements, and are characterised by signatures of positive evolutionary selection[2]. Tumour genomes also carry numerous passenger mutations, which do not impact cell phenotype and are evolutionarily neutral, yet typically outnumber drivers[3]. Identification of driver mutations, and the "driver genes" through which they act, is a critical step towards understanding and treating cancer[1,4]. Computational driver gene discovery tools continue to be refined, yielding catalogues of increasing accuracy that form the foundation of precision therapeutic development[4,5]. Driver genes represent a subset of more broadly defined "cancer genes", the latter defined as those that functionally promote or oppose oncogenic cell states, regardless of mutational status[6].

Most tumour types are characterised by a limited and recurrent sequence of driver mutations, which promote disease hallmarks via functional changes to encoded oncogene or tumour suppressor proteins[6,7]. However, the vast majority of somatic single-nucleotide variants (SNVs) fall outside protein-coding genes[8]. Combined with increasing awareness of the disease roles of noncoding genomic elements[9], this raises the question of whether non-protein-coding mutations also shape cancer cell fitness[10]. Growing numbers of both theoretical[11–16] and experimental studies[2,17–20] have linked noncoding SNVs to cell fitness through alterations in the function of elements such as enhancers, promoters, insulator elements and small RNAs[10,21].

One particularly important class of cancer-promoting non-protein-coding elements are the long noncoding RNAs (lncRNAs)[22]. LncRNA transcripts are modular assemblages of functional elements that can interact with other nucleic acids and proteins via defined sequences or structural elements[23,24]. Of the >50,000 loci mapped in the human genome[25], hundreds of "cancer-lncRNAs" have been demonstrated to act as oncogenes or tumour suppressors[26] via up- or downregulated expression in tumours. Their clinical importance is further supported by prognosis[27], copy number variants (CNVs)[28–30], tumour-initiating transposon screens in mouse[31] and function-altering germline cancer variants[32].

LncRNA genes also tend to be highly mutated in tumour DNA[2,33,34]. For example, the *NEAT1* lncRNA, which is a structural component of subnuclear paraspeckle bodies, has been noted for its high mutation rate across a variety of cancers[2,33,35]. This raises the possibility that a subset of cancer lncRNAs may also act as "driver-lncRNAs", where SNVs promote cell fitness by altering lncRNA activity. However, recent studies have argued that mutations in *NEAT1* and other lncRNAs arise from phenotypically neutral passenger effects[2,33]. To date, the fitness effects of lncRNA SNVs have not been investigated experimentally, leaving the existence of driver lncRNAs unresolved.

In this study, we investigate the existence of driver lncRNAs. We develop an enhanced lncRNA driver discovery pipeline, and use it to comprehensively map candidate driver lncRNAs using somatic SNVs from thousands of primary and metastatic tumours. We evaluate the clinical and genomic properties of these candidates. Finally, we employ a range of functional and mechanistic assays to gather experimental evidence for fitness-altering driver mutations acting through lncRNAs.

## Results

### Integrative driver lncRNA discovery with ExInAtor2
Driver genes can be identified by signals of positive selection acting on their somatic mutations. The two principal signals are *mutational burden* (MB), an elevated mutation rate, and *functional impact* (FI), the degree to which mutations are predicted to alter encoded function. Both signals must be compared to an appropriate background, representing mutations under neutral selection.

To search for lncRNAs with evidence of driver activity, we developed *ExInAtor2*, a driver discovery pipeline with enhanced sensitivity due to two key innovations: integration of both MB and FI signals, and empirical background estimation (see "Methods") (Fig. 1a and Supplementary Fig. S1a, b). For MB, local background rates are estimated, controlling for covariates of mutational signatures and large-scale effects such as replication timing, which otherwise can confound driver gene discovery[36]. For FI, we adopted functionality scores from the *Combined Annotation Dependent Depletion* (CADD) system, due to its widespread use and compatibility with a range of gene biotypes[37]. Importantly, *ExInAtor2* remains agnostic to the biotype of genes/functional elements, allowing independent benchmarking with established protein-coding gene data.

### Discovery of lncRNA and protein-coding driver genes
We began by benchmarking ExInAtor2 using the maps of somatic single-nucleotide variants (SNVs) from tumour genomes sequenced by the recent PanCancer Analysis of Whole Genomes (PCAWG) project[1], comprising altogether 45,704,055 SNVs from 2583 donors (Fig. 1b and "Methods"). As it was generated from whole-genome sequencing (WGS), this dataset makes it possible to search for driver genes amongst both non-protein-coding genes (including lncRNAs) and better-characterised protein-coding genes.

To maximise sensitivity and specificity, we prepared a carefully filtered annotation of lncRNAs. Beginning with high-quality curations from GENCODE[38], we isolated intergenic lncRNAs lacking evidence for protein-coding capacity. To the resulting set of 6981 genes (Fig. 1c), we added the set of 294 confident, literature-curated lncRNAs from the Cancer LncRNA Census 2 resource[26], for a total set of 7275 genes.

We compared the performance of ExInAtor2 to ten leading driver discovery methods and PCAWG's consensus measure (PCAWGc), which integrates and outperforms all individual methods (Fig. 2a)[2]. Performance was benchmarked on curated sets of protein-coding and lncRNA cancer genes (Fig. 2b). Judged by correct identification of cancer lncRNAs at a false discovery rate (FDR) cutoff of <0.1, ExInAtor2 displayed the best overall accuracy in terms of $F_1$ measure (Fig. 2c, d). Quantile–quantile (QQ) analysis of resulting $P$ values (P) displayed no obvious inflation or deflation and has amongst the lowest mean log-fold change (MLFC) values (Fig. 2e), together supporting ExInAtor2's low and controlled FDR.

ExInAtor2 is biotype-agnostic, and protein-coding driver datasets are highly refined (Fig. 2b). To further examine its performance, we evaluated sensitivity for known protein-coding drivers from the benchmark Cancer Gene Census[39]. Again, ExInAtor2 displayed competitive performance, characterised by low false-positive predictions (Supplementary Fig. S2a–c).

To test ExInAtor2's FDR estimation, we repeated the lncRNA analysis on a set of carefully randomised pan-cancer SNVs (see "Methods"). Reassuringly, no hits were discovered, and QQ plots displayed neutral behaviour (MLFC 0.08) (Supplementary Fig. S2d). Analysing at the level of individual cohorts, ExInAtor2 predicted 3/40 lncRNA-cohort associations in the simulated/real datasets, respectively. This corresponds to an empirical FDR rate of 0.075, consistent with the nominal FDR cutoff of 0.1.

We conclude that ExInAtor2 identifies known driver genes with a low and controlled false discovery rate.

### The landscape of driver lncRNA in primary human tumours
We next set out to create a genome-wide panorama of mutated lncRNAs across human primary cancers. Tumours from PCAWG were grouped into a total of 37 cohorts, ranging in size from two tumours (Cervix-AdenoCa, Lymph-NOS and Myeloid-MDS tumour types) to 314 (Liver-HCC tumour type), in addition to the entire pan-cancer set (Fig. 3a).

After removing likely false-positive associations using the same stringent criteria as PCAWG[1], ExInAtor2 revealed altogether 21 unique cancer-lncRNA associations, involving 17 lncRNAs (Fig. 3b)—henceforth considered putative driver lncRNAs. Of these, nine are annotated lncRNAs that have not previously been linked to cancer, denoted "novel". The remaining "known" candidates are identified in the literature-curated Cancer LncRNA Census 2 dataset[26]. Known lncRNAs tend to be hits in more individual cohorts than novel lncRNAs, with cases like *NEAT1* being detected in four cohorts (Fig. 3b). While most driver lncRNAs display exonic mutation rates ~50-fold greater than background (coloured cells, Fig. 3b), the number of mutations in such genes is diverse between cohorts, being Pancancer, Lymph-CLL and Skin-Melanoma the biggest contributors of mutations.

Supporting the accuracy of these predictions, the set of driver lncRNAs is highly enriched for known cancer lncRNAs[26] (8/17 or 48%, Fisher test $P = 2e$-6) (Fig. 3c). Driver lncRNAs are also significantly enriched in three other independent literature-curated databases (Supplementary Fig. S3a).

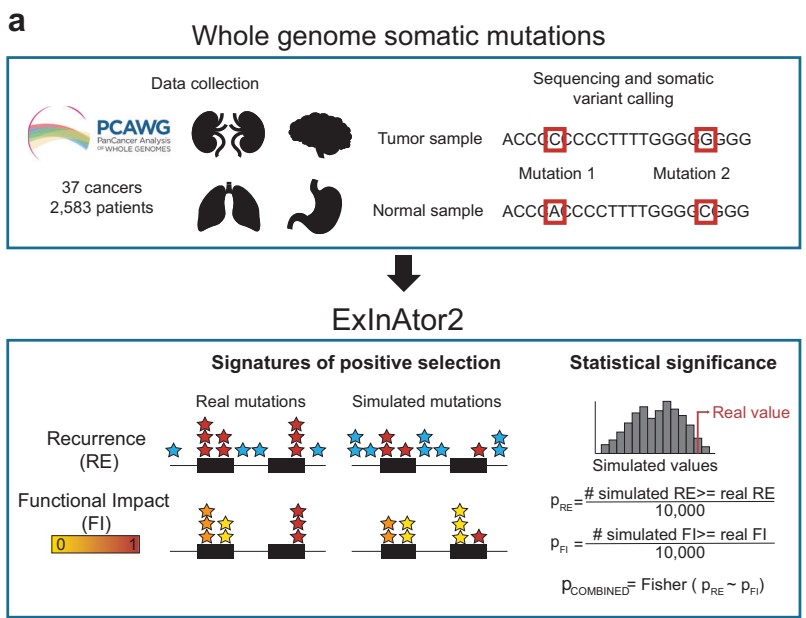

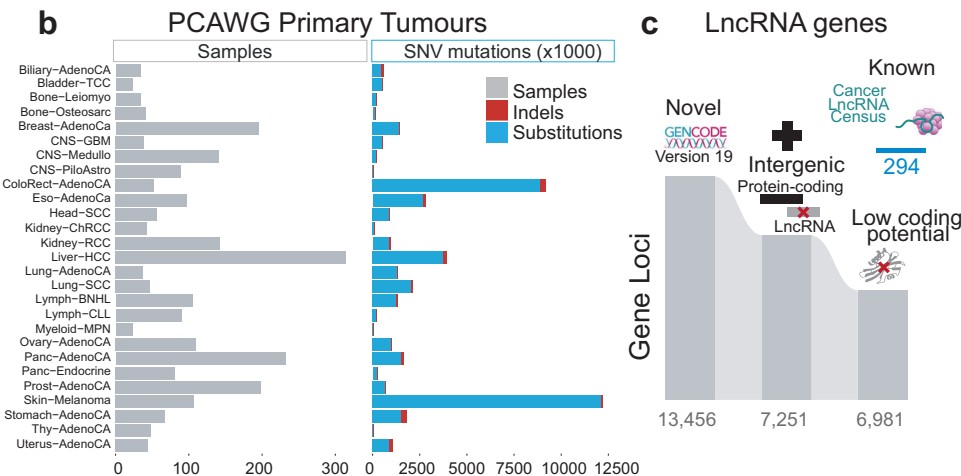

**Fig. 1 | Driver lncRNA discovery with ExInAtor2. a** ExInAtor2 accepts input in the form of maps of single-nucleotide variants (SNVs) from cohorts of tumour genomes. Two signatures of positive selection are evaluated, and compared to simulated local background distributions to evaluate statistical significance. The two significance estimates are combined using Fisher's method. **b** Summary of the primary tumour datasets used here, obtained from Pancancer Analysis of Whole Genomes (PCAWG) project. **c** A filtered lncRNA gene annotation was prepared, and combined with a set of curated cancer lncRNAs from the Cancer LncRNA Census[26].

We also searched for evidence of epistatic interactions between SNVs in lncRNA drivers and other lncRNA or known protein-coding drivers. Although we could retrieve many known PCG–PCG interactions, both positive and negative, we found no example of an lncRNA SNV participating in such an interaction (Supplementary Data 7 and 8).

**Driver lncRNAs carry features of functionality and clinical relevance**

To further evaluate the quality of driver lncRNA predictions, we tested their association with genomic and clinical features expected of bona fide cancer genes, defined as those validated by functional assays in vitro and in vivo from the scientific literature[26]. LncRNA catalogues are likely to contain a mixture of both functional and non-functional genes. The former group is characterised by purifying evolutionary selection and high expression in healthy and diseased tissues[31]. We found that driver lncRNAs display higher evolutionary sequence conservation and higher steady-state levels in healthy organs (Fig. 3d).

Their sequence also contains more microRNA binding sites, suggesting integration with post-transcriptional regulatory networks.

In contrast, we could find no evidence that driver lncRNAs are enriched for genomic covariates and features arising from artefactual results. They have earlier replication timing (whereas later replication is associated with greater passenger mutation rates)[40], less exonic repetitive sequence (ruling out mappability biases), and similar exonic GC content (ruling out sequencing bias) compared to tested non-candidates (Fig. 3d). However, driver lncRNAs tend to have longer spliced length, likely reflecting greater statistical power for longer genes that affects all driver methods[33].

Driver lncRNAs also have clinical features of cancer genes (Fig. 3e). They are on average 158-fold higher expressed in tumours compared to normal tissues (133 vs 0.84 FPKM) (Fig. 3e, PCAWG RPKM), 2.15-fold enriched for germline cancer-associated small nucleotide polymorphism (SNP) in their gene body (4.7% vs 2.5%) (Fig. 3e, SNPs per MB), and enriched in orthologues of driver lncRNAs carrying common insertion sites (CIS), discovered by transposon insertional mutagenesis (TIM)

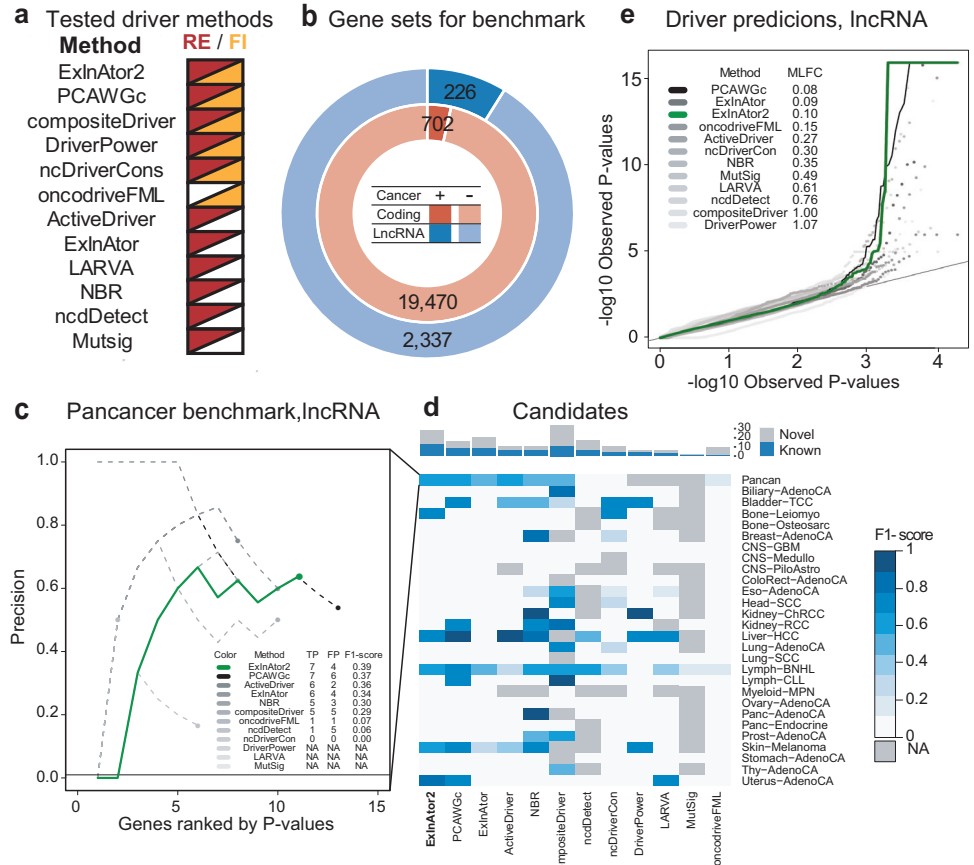

**Fig. 2 | ExInAtor2 accurately identifies driver genes. a** The list of driver discovery methods to which ExInAtor2 was compared. The signatures of positive selection employed by each method are indicated to the right. PCAWGc indicates the combined driver prediction method from Pan-Cancer Analysis of Whole Genomes (PCAWG), which integrates all ten methods. **b** Benchmark gene sets. LncRNAs (blue) were divided in positives and negatives according to their presence or not in the Cancer LncRNA Census[26], respectively, and similarly for protein-coding genes in the Cancer Gene Census[39]. **c** Comparing performance in terms of precision in identifying true positive known cancer lncRNAs from the CLC dataset, using PCAWG Pancancer cohort. *x* axis: genes sorted by increasing *P* value (uncorrected for multiple hypothesis testing) from each correspondent method, as described in ref. [2]. *y* axis: precision, being the percentage of true positives amongst cumulative set of candidates at increasing *P* value cutoffs. The horizontal line shows the baseline, being the percentage of positives in the whole list of tested genes.

Coloured dots represent the precision at cutoff of $q \le 0.1$ (Benjamini–Hochberg method). Inset: Performance statistics for cutoff of $q \le 0.1$. **d** Driver prediction performance for all methods in all PCAWG cohorts. Cells show the F1-score of each driver method (*x* axis) in each cohort (*y* axis). Grey cells correspond to cohorts where the method was not run. The bar plot on the top indicates the total, non-redundant number of True Positives (TP) and False Positives (FP) calls by each method. Driver methods are sorted from left to right according to the F1-score of unique candidates. **e** Evaluation of *P* value distributions for driver lncRNA predictions. Quantile–quantile plot (QQ-plot) shows the distribution of observed vs expected −log10 *P* values (uncorrected for multiple hypothesis testing) for each method run on the PCAWG Pancancer cohort (uncorrected for multiple hypothesis testing; as described in ref. [2]). The Mean Log-Fold Change (MLFC) quantifies the difference between observed and expected values ("Methods").

screens in mouse (17.6% vs 1.6%) (Supplementary Fig. S3a, Transposon insertion mutagenesis)[26]. Finally, driver lncRNAs significantly overlap growth-promoting hits discovered by CRISPR functional screens (11.8 vs 1.3%) (Supplementary Fig. S3a, Growth-promoting). In conclusion, driver lncRNA display evidence for functionality across a wide range of functional and clinical features, strongly suggesting that they are enriched for bona fide cancer-driver genes.

**The landscape of driver lncRNAs in metastatic tumours**
We further extended the driver-lncRNA landscape to metastatic tumours, using 3527 genomes from 31 cohorts sequenced by the Hartwig Medical Foundation (Supplementary Fig. S3b–d)[41]. Performing a similar analysis as above, we identified 43 driver lncRNAs in a total of 53 lncRNA-tumour combinations (Supplementary Fig. S3c). Eight predicted drivers are known cancer lncRNAs, significantly higher than random expectation (*P* = 0.004, Fisher exact test) (Fig. 3c). Further adding confidence to these findings is the significant overlap of driver lncRNAs from metastatic and primary tumour cohorts (Fig. 3c).

**Driver mutations identify oncogenic lncRNAs**
We wished to evaluate the functional disease relevance of driver lncRNAs, and particularly those that had not previously been implicated in cancer. Thus, we overexpressed a panel of nine candidates in HeLa cells and found that three promote cell growth (Fig. 4a and Supplementary Fig. S4a).

It was interesting to note that, amongst the six that did not display a significant effect was a lncRNA, AC087463.1, herein named *LOHAN1* (LncRNA Oncogene in Head and Neck cancer−ENST00000568541 at the PWRN1 locus), which appeared as a potential driver in the Head and Neck (HN) tumour cohort (Fig. 3b). *LOHAN1* is transcribed from the same locus as the lncRNA *PWRN1*, previously reported as a tumour suppressor in gastric cancer[42]. The overexpression of *LOHAN1*, as well as of another driver lncRNA, *LOHAN2* (LncRNA Oncogene in Head and Neck cancer−ENSG00000258779.2; RP11-140I24), increased tumorigenicity in head and neck (HN) cells, as measured by colony-forming potential (Fig. 4b, c), supporting the notion that some lncRNAs have a cell-type-specific activity.

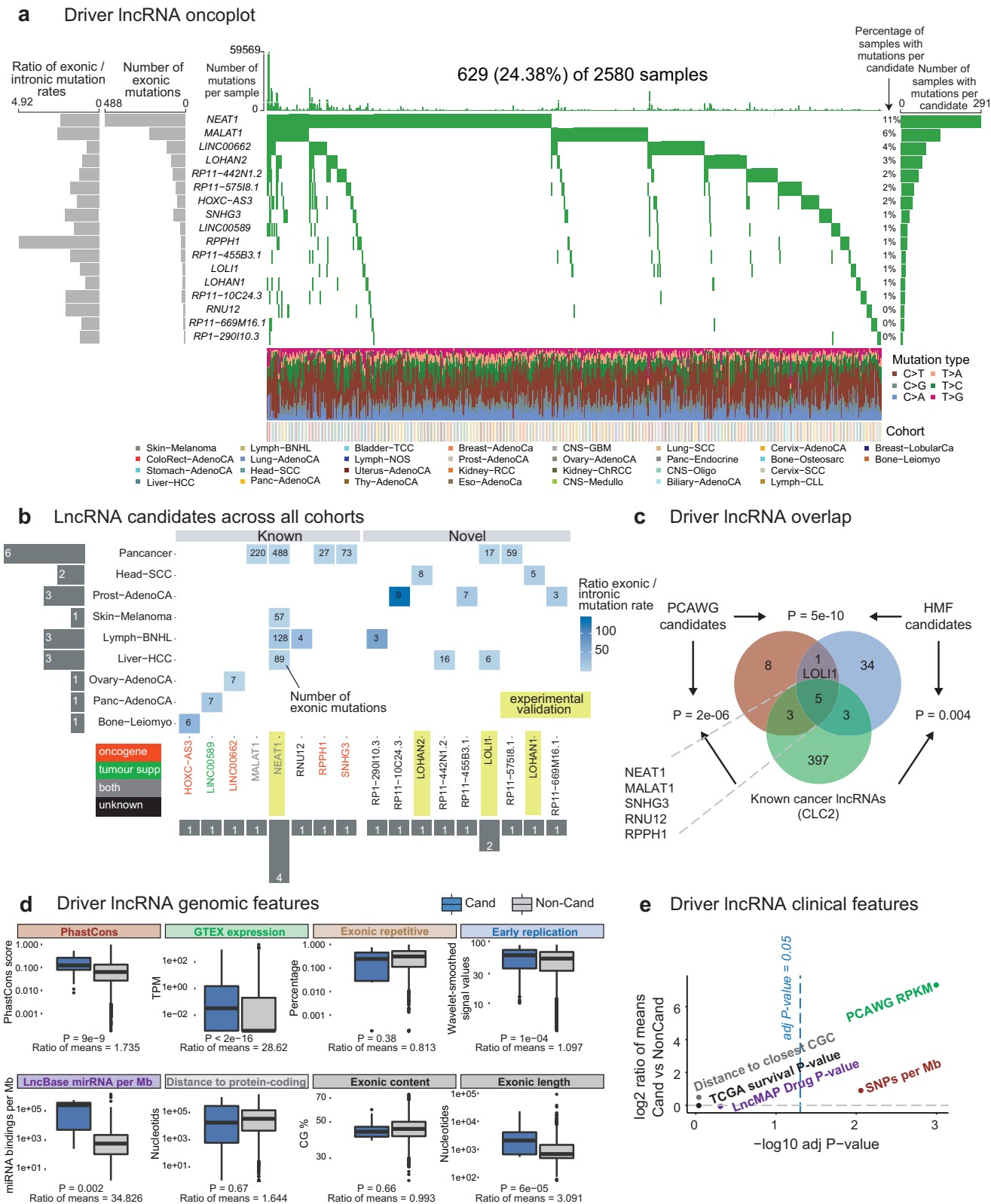

**a** Driver lncRNA oncoplot

**b** LncRNA candidates across all cohorts

**c** Driver lncRNA overlap

**d** Driver lncRNA genomic features

**e** Driver lncRNA clinical features

ENSG00000241219 (RP11-572M11.1), herein named *LOLI1* (LncRNA Oncogene in Liver cancer 1) displayed elevated mutation rates in Hepatocellular Carcinoma (HCC) tumours (Figs. 3b and 4d) and was detected as driver in both the PCAWG and HFM datasets (Fig. 3c). We could not find any studies on this lncRNA in prior scientific literature. According to the latest GENCODE version 38, its single-annotated isoform comprises three exons, and displays low expression in normal tissues (Supplementary Fig. S4b). We could detect *LOLI1* in two HCC cell lines, HuH7 and SNU-475 (Fig. 4f and Supplementary Fig. S4d). To perturb *LOLI1* expression, we designed two different antisense oligo-nucleotides (ASOs) that reduced steady-state levels by >50% in both cell lines (Fig. 4e, f and Supplementary Fig. S4d). We evaluated the role of *LOLI1* in HCC cell proliferation, by measuring changes in growth rates following ASO transfection. The significant decrease in growth

**Fig. 3 | The landscape of driver lncRNAs in primary tumours. a** "Oncoplot" overview of driver lncRNA analysis in PCAWG primary tumours. Rows: 17 candidate driver lncRNAs at cutoff of FDR ≤ 0.1. Columns: 2580 tumours. **b** LncRNA candidates across all cohorts. Rows: Cohorts where hits were identified. Columns: 17 candidate driver lncRNAs. "Known" lncRNAs are part of the literature-curated Cancer LncRNA Census (CLC2) dataset[26]. Functional labels (oncogene/tumour suppressor/both) were also obtained from the same source. **c** Intersection of candidate driver lncRNAs identified in PCAWG primary tumours, Hartwig Medical Foundation (HMF) metastatic tumours and the CLC2 published cancer-lncRNA set. Statistical significance was estimated by Fisher's exact test. **d** Genomic features of driver lncRNAs. Each plot displays the values of indicated features for 17 candidate driver lncRNAs (blue) and all remaining tested lncRNAs (non-candidates, grey).

Significance was calculated using two-sided Wilcoxon test, (uncorrected for multiple hypothesis testing). For each comparison, the ratio of means was calculated as (mean of candidate values/mean of non-candidate values). Centre = medians (Line), bottom and top boundaries of the box = 25 and 75th percentiles of the data, minima and maxima = lowest and highest data points. See "Methods" for more details. **e** Clinical features of driver lncRNAs. Each point represents the indicated feature. *y* axis: log2-transformed ratio of the mean candidate value and mean non-candidate value. *x* axis: The statistical significance of candidate vs non-candidate values, as estimated by a two-sided Wilcoxon test and corrected for multiple testing with Benjamini–Hochberg method. See "Methods" for more details. Source data are provided as a Source Data file.

resulting from both ASOs in both cell backgrounds points to the importance of *LOLI1* in HCC cell fitness (Fig. 4g and Supplementary Fig. S4e).

These results prompted us to ask whether *LOLI1* can also promote cell growth in other cancer types. Thus, we turned to CRISPR activation, to upregulate the lncRNA from its endogenous locus in HeLa cervical carcinoma cells. Three independent sgRNAs increased gene expression by 4 to ~20-fold (Fig. 4h and Supplementary Fig. S4c), of which two significantly and specifically increased cell proliferation (Fig. 4i).

Having established that *LOLI1* promotes cell growth in a number of backgrounds, we next asked whether tumour mutations can enhance this activity, as would be expected for driver mutations. To do so, we designed and validated overexpression plasmids for the wild-type or mutated forms of the transcript (Fig. 4j). We first tested mutations from two individual patients separately. We selected mutations that were recurrently observed in independent tumours from both PCAWG and HFM datasets (Fig. 4d) (i.e., Mut1, including two mutations identified within the same patient and Mut2, including three mutations from the same patient, grey boxes, depicted in Supplementary Fig. S4g). Importantly, transfection of wild-type *LOLI1* boosted cell growth, consistent with ASO results above (Supplementary Fig. S4f, WT). However, neither of the individual patients' mutations alone yielded statistically significant changes in cell growth (Supplementary Fig. S4f). We hypothesised that our experimental model may be too insensitive to detect subtle changes from individually weak mutations. Therefore, we combined the four SNVs from both patients and observed a significant additional increase in cell proliferation compared to wild-type *LOLI1* (Fig. 4k). These results were further corroborated in a non-transformed immortalised human hepatocyte (IHH) background, where mutant *LOLI1* similarly boosts cell viability (Fig. 4l, m). Finally, we obtained primary hepatocytes from a human healthy donor and observed that plasmid-mediated expression of mutated *LOLI1* promoted the upregulation of proliferation-associated cytokines compared to untreated and control-transduced hepatocytes (Fig. 4n). In summary, individual recurrent tumour SNVs in *LOLI1* have a relatively weak effect, consistent with the Weak Driver hypothesis[43], yet in combination are sufficient to produce significant increases in cell viability in both transformed and non-transformed backgrounds.

### Mutations in *NEAT1* promote cell fitness and correlate with survival

To gain mechanistic insights into how fitness-enhancing driver mutations may act through lncRNAs, we turned to *NEAT1* a relatively well-understood lncRNA for which confident mechanistic and functional data is available[44,45]. Based on ExInAtor2 analysis, *NEAT1* mutations, spanning the entire gene length, display evidence for positive selection in altogether 4 and 3 cancer cohorts in PCAWG and HMF datasets, respectively. PCAWG and others also noted this highly elevated mutation rate in the *NEAT1* gene, although it has been argued that these result from neutral passenger processes, possibly linked to the high expression of the gene[2,35,46].

*NEAT1* produces short and long isoforms (called *NEAT1_1* / *NEAT1_2*) of 3.7 and 22.7 kb, respectively[47], which are completely overlapping at the 5' of the gene (Fig. 5b). *NEAT1_1* is a ubiquitous, abundant, polyadenylated and highly conserved transcript[48]. In contrast, *NEAT1_2*, responsible for the formation of membraneless nuclear paraspeckle structures, is not polyadenylated and expressed under specific conditions or in response to various forms of stress[49,50].

We sought to test whether SNVs in *NEAT1* can act as drivers. We hypothesised that tumour SNVs could be simulated by wild-type Cas9 protein, which is known to cause similar mutations when double-strand breaks are resolved by error-prone DNA repair[18,51]. We selected six regions of *NEAT1*, based on high mutation density, evolutionary conservation and known functions[52], hereafter called Reg1, Reg2, etc., and targeted them with altogether 15 sgRNAs (Fig. 5a, b). To control for the non-specific fitness effects of double-strand breaks (DSBs)[53,54], we also created two neutral control sgRNAs targeting *AAVS1* locus, and an efficient positive-control-paired sgRNA (pgRNA) to delete the entire *NEAT1_1* region ("KO-sgRNA" in Fig. 5b and Supplementary Fig. S5a). Sequencing of treated cells' gDNA revealed narrowly focussed substitutions and indels at target regions, similar to that observed in real tumours (Fig. 5c and Supplementary Fig. S5b).

To quantify mutations' effects on cell fitness, we established a competition assay between mutated mCherry-labelled cells and control GFP-labelled cells (Supplementary Fig. 4h, Fig. 5d and Supplementary Fig. S5c)[18]. As expected, deletion of entire *NEAT1_1* (knockout, KO) in HeLa cells led to reduced growth, while control sgRNAs did not (Fig. 5d). Notably, HeLa cells carrying *NEAT1* mutations in defined regions displayed increased fitness: two at the 5' of the gene (Reg2 and Reg3), one internally near the alternative polyadenylation site (Reg4) and one at the 3' end (Reg5) (blue line, Fig. 5d and Supplementary Fig. S5c). These findings were supported in 3/4 cases in HCT116 colorectal carcinoma cells (green line, Fig. 5d and Supplementary Fig. S5c). The effect of NEAT1-targeting sgRNAs was lost in a *NEAT1* KO background, indicating that changes in cell fitness arise as a result of on-target *NEAT1* mutations (Supplementary Fig. S5d).

To corroborate these findings, we repeated fitness assays in a more complex pooled competition assay. Here, the evolution of defined mixtures of mutant cells is quantified by amplicon sequencing of sgRNA barcodes. Consistent with previous results, cells carrying *NEAT1* mutations outcompeted control cells over time (Fig. 5e).

These results were obtained from monolayer cells, whose relevance to real tumours is disputed. Thus, we performed additional experiments in 3-dimensional spheroids grown from mutated HCT116 cells, and observed again that Reg2 mutations led to increased growth (Fig. 5f).

The experiments thus far were performed in transformed cancer cells. To investigate whether *NEAT1* mutations also enhance fitness in a non-transformed background, we performed similar experiments in MRC5 immortalised foetal lung fibroblasts. Again, *NEAT1* mutations were observed to increase fitness, in terms of cell growth (Fig. 5g) and, at least for Reg2, in terms of anchorage-independent growth (Fig. 5h).

To further test *NEAT1* mutations' driver potential in a realistic in vivo setting, we turned to a widely used nude mouse model. We

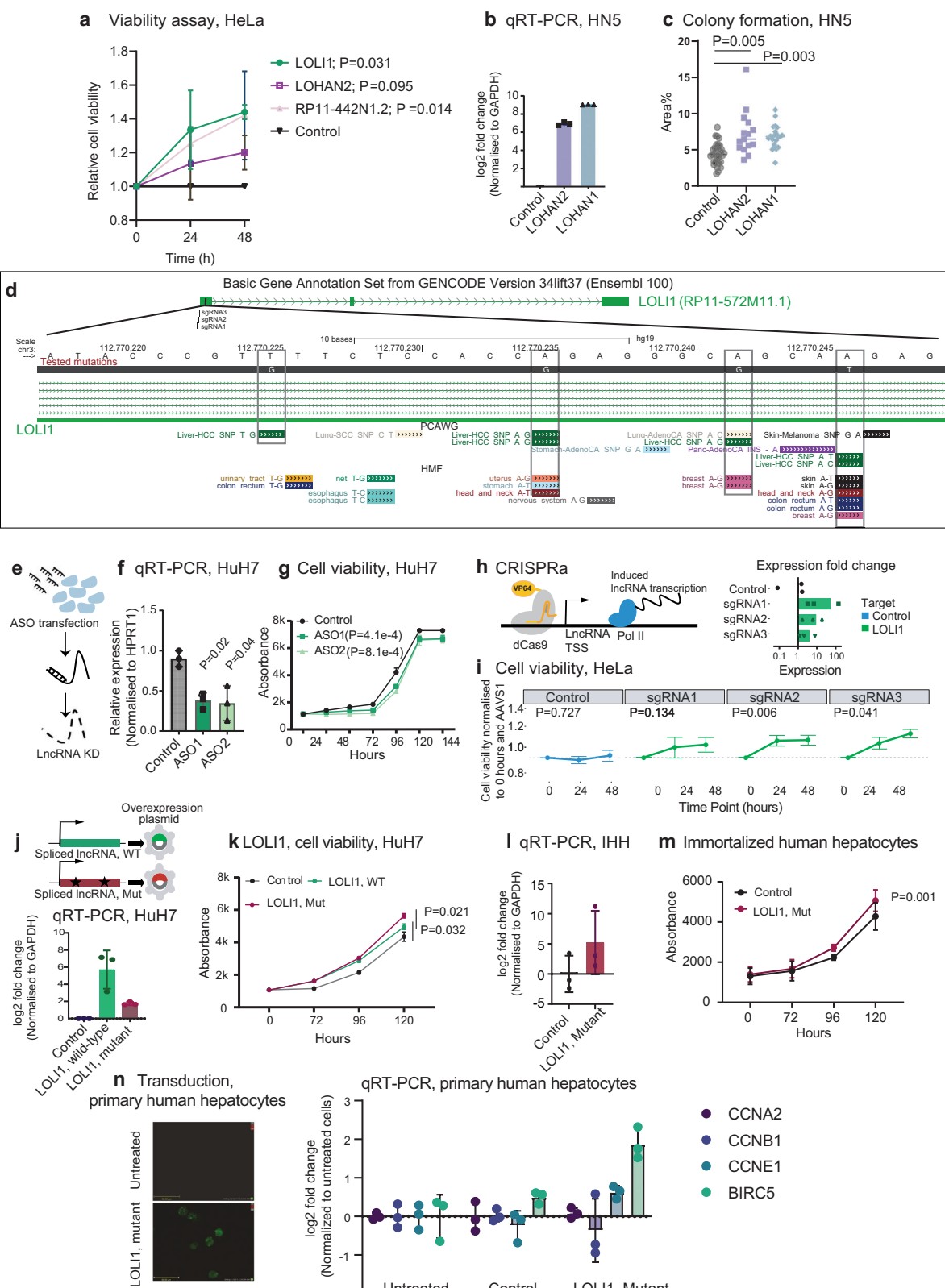

**a** Viability assay, HeLa

**b** qRT-PCR, HN5

**c** Colony formation, HN5

**d** Basic Gene Annotation Set from GENCODE Version 34lift37 (Ensembl 100)

**e** ASO transfection

**f** qRT-PCR, HuH7

**g** Cell viability, HuH7

**h** CRISPRa

**i** Cell viability, HeLa

**j** Overexpression plasmid / qRT-PCR, HuH7

**k** LOLI1, cell viability, HuH7

**l** qRT-PCR, IHH

**m** Immortalized human hepatocytes

**n** Transduction, primary human hepatocytes / qRT-PCR, primary human hepatocytes

implanted HCT116 cells carrying Reg2 and Reg3 mutations subcutaneously, and assayed the growth of resulting mutations (Fig. 5i). After 21 days, Reg2-mutant tumours were significantly larger than controls, providing the strongest support yet that mutations in *NEAT1* at the Reg2 position have fitness-altering driver activity.

Finally, to test their relevance in human cancer patients, we asked whether the presence of a *NEAT1* mutation correlates with survival.

Indeed, in lymphoid cancer patients from the PCAWG cohort, *NEAT1* mutations correlate with significantly worse prognosis (Fig. 5j). This effect remains even after accounting for differences in total mutation rates using the Cox proportional hazards model ($P = 0.02$).

In summary, *NEAT1* tumour mutations consistently increase cell fitness in vitro and in vivo, in a range of genetic backgrounds, and are associated with poor prognosis in lymphoid cancer patients.

**Fig. 4 | Functional effects of driver lncRNAs in cell viability. a** Plasmid-transfected cells were measured at indicated timepoints. Statistical significance was estimated by two-sided Student's *t* test based on *n* = 3 independent replicates. Mean value +/− SD is plotted. Replicates were performed at different times (experimental replicates). **b** Overexpression of *LOHAN1&2* in HN5 cells. RNA levels were measured by qRT-PCR; *n* = 3 (experimental replicates performed at different times). **c** Results of colony formation assay in HN5 cells. Data are presented as mean values −/+ SD of the percent of well area covered from 18 culture wells. Statistical significance was estimated using one-way ANOVA. Replicates were performed at the same time (technical replicates). **d** The genomic locus of the lncRNA *LOLI1*. Also shown are SNVs from PCAWG and HMF cohorts. The SNVs included in the mutated plasmid are indicated in the grey boxes. **e** ASOs were transfected to knock down *LOLI1* expression and **f** RNA levels measured in HuH7 cells. Statistical significance was estimated using one-sided Student's *t* test; *n* = 3 (experimental replicates). **g** ASO-transfected cells were measured at indicated timepoints; *n* = 3. Statistical significance was estimated by linear regression model on log2 value (experimental replicates). **h** CRISPRa targeting *LOLI1*. On the right, qRT-PCR measurements of

*LOLI1* with indicated sgRNAs; *n* = 3 (experimental replicates). **i** The effect of CRIS-PRa on HeLa cells' viability; *n* = 6 (experimental replicates). Statistical significance was estimated by one-sided paired *t* test at 48 h. **j** Plasmids expressing spliced *LOLI1* sequence, in wild-type (WT) or mutated form (Mut) were transfected into HuH7 cells. RNA levels were measured by qRT-PCR; *n* = 3 (experimental replicates). **k** Populations of plasmid-transfected cells were measured at indicated timepoints. Statistical significance was estimated by one-sided Student's *t* test based; *n* = 3 (experimental replicates). **l** *LOLI1* overexpression in immortalised human hepatocytes (IHH). RNA levels were measured by qRT-PCR; *n* = 3 (experimental replicates). **m** The viability of IHH was measured at indicated timepoints. Statistical significance was calculated by one-sided Student's *t* test; *n* = 3 (experimental replicates). **n** Primary human hepatocytes were transduced to overexpress *LOLI1* mutant transcript (left panel). Transduction was monitored by EGFP marker gene (left panel). The change in proliferation-associated cytokine was measured by qRT-PCR (right panel); *n* = 6 experimental replicates. *n* indicates the number of independent experiments. Data show the mean value +/− SD in (**a**–**c**, **f**–**n**). Source data are provided as a Source Data file.

## Mutations alter the *NEAT1* protein interactome and increase paraspeckle formation

*NEAT1* is a necessary component of subnuclear paraspeckles, which assemble when specific architectural proteins bind to nascent NEAT1_2 transcripts[55]. Paraspeckles are nuclear condensates containing diverse gene-regulatory proteins[49]. They are often observed in cancer cells[56], and are associated with poor prognosis[57]. Thus, we hypothesised that *NEAT1* mutations might affect cell fitness via alterations in paraspeckle number or structure.

We first checked for possible changes in *NEAT1* expression and isoform usage in response to experimentally-delivered mutations. Mutations caused no statistically significant change in *NEAT1_1* expression, while deletion of *NEAT1_1* (KO) reduced steady-state levels, as expected (Supplementary Fig. S5f). Reassuringly, the only mutation to significantly increase *NEAT1_2* levels was in Region 4 (Supplementary Fig. S5g), which is consistent with the fact that it contains the alternative polyadenylation site that mediates switching between the short- and long isoforms[58].

Using fluorescence in situ hybridisation (FISH) with *NEAT1_2* probes, we next asked whether mutations impact on paraspeckle number or size (Fig. 6a–c). Despite changes in isoform expression noted above, mutations in Region 4 resulted in no change in the number or size of paraspeckles, in line with previous findings[52] (Fig. 6a, b). However, mutations in Region 2 yielded a significant increase in the number and size of paraspeckles (Fig. 6a–c). Thus, SNVs in *NEAT1* can impact paraspeckles.

*NEAT1* is known to function via a diverse cast of protein partners. Region 2 mutations overlap several known protein binding sites, and fall in or near to areas of deep evolutionary conservation of sequence and structure (Supplementary Fig. S5h).

To better understand how Region 2 mutations alter *NEAT1* function, and evaluate if mutation could affect the binding of proteins to *NEAT1*, we compared the protein interactome of wild-type and mutant RNA by in vitro pulldown coupled to mass spectrometry (Fig. 6d). We created a 288 nt fragment of NEAT1–Region 2 for wild-type (WT) and mutated (MUT) sequence, the latter containing two SNVs observed in patient tumours (Fig. 6d). We performed RNA pulldown with nuclear lysates from HeLa cells, followed by mass spectrometry. Altogether, 154 interacting nuclear proteins were identified for wild-type sequence. Supporting the usefulness of this approach, interacting proteins highly enriched for both known NEAT1-binders and paraspeckle proteins (see "Methods") and include well known examples like NONO[52,59] (Fig. 6e). Comparing mutant to wild-type interactomes, we observed widespread changes in *NEAT1* complexes: altogether 8 (4.6%) proteins are lost by mutant RNA, and 18 (10.3%) gained (Fig. 6f). STRING analysis revealed that the proteins lost upon mutation are highly enriched for members of the core RNA Polymerase II complex

(strength = 2.51, *P* = 0.016; basic list enrichment by STRING, Benjamini–Hochberg corrected) and physically interacting with other proteins of this complex (Fig. 6g). The accuracy of reported changes in protein binding was supported by independent RNA immunoprecipitation experiments using antibodies for two differentially bound proteins, PQBP1 and SREK1 (Fig. 6h and Supplementary Fig. S5i, j)[60].

*NEAT1*-interacting proteins are expected to play roles in paraspeckle formation. To test this, we knocked down *NEAT1* interactors identified here and tested the effect on paraspeckle formation and cell viability. Intriguingly, we observed a striking phenotype from U2SURP knockdown, with the formation of elongated paraspeckle structures (Fig. 6i, j), indicating U2SURP to be a key paraspeckle protein[60]. In addition, a decrease in cell proliferation was observed, in line with previous observations[61]. For SREK1, whose interaction is lost in mutant *NEAT1*, we found that loss of function led to increased paraspeckle counts (Fig. 6j, k) and cell proliferation (Fig. 6l). We investigated whether mutations create or destroy known binding motifs of changing proteins, but could find no evidence for this. Altogether, these findings suggest a model where tumour SNVs alter the protein interactome of *NEAT1*, leading to both gains and losses of protein partners. For example, SREK1 appears to bind wild-type *NEAT1* to repress formation of paraspeckles, and this interaction in abrogated by Region 2 mutations to boost paraspeckles and consequently accelerate cell proliferation (Fig. 6m). It is likely that the gain and loss of other protein partners also contribute to mutation-associated changes in paraspeckle numbers and form.

## Discussion

Understanding which tumour mutations are drivers that promote pathogenic cell fitness, and how they do so, are fundamental goals of cancer genomics. Here we have focussed on a particularly intriguing class of potential driver genes, the lncRNAs, which are known to be both potent cancer genes and highly mutated in tumours, and yet for which no driver mutation has been experimentally validated to date[2,33,35,62].

To address this gap, we here developed an improved method, ExInAtor2, to search for driver lncRNAs based on integrated signatures of positive selection. ExInAtor2 is straightforward to run by bioinformaticians or Unix-literate biologists, is competitive with present-day tools, and is freely available in Github. In total, we identified 54 candidate driver lncRNAs across an extensive tumour cohort, including both primary and metastatic tumours. The value of these predictions is supported by consistency between independent cohorts, overlap with various cancer-lncRNA databases, and from functional screens in mouse. Nevertheless, in silico driver analyses suffer from a variety of constraints, from false positives due to localised, non-selected mutational processes, to false negatives due to the limited sample size. Such factors have limited the confidence with which previous studies[33,34]

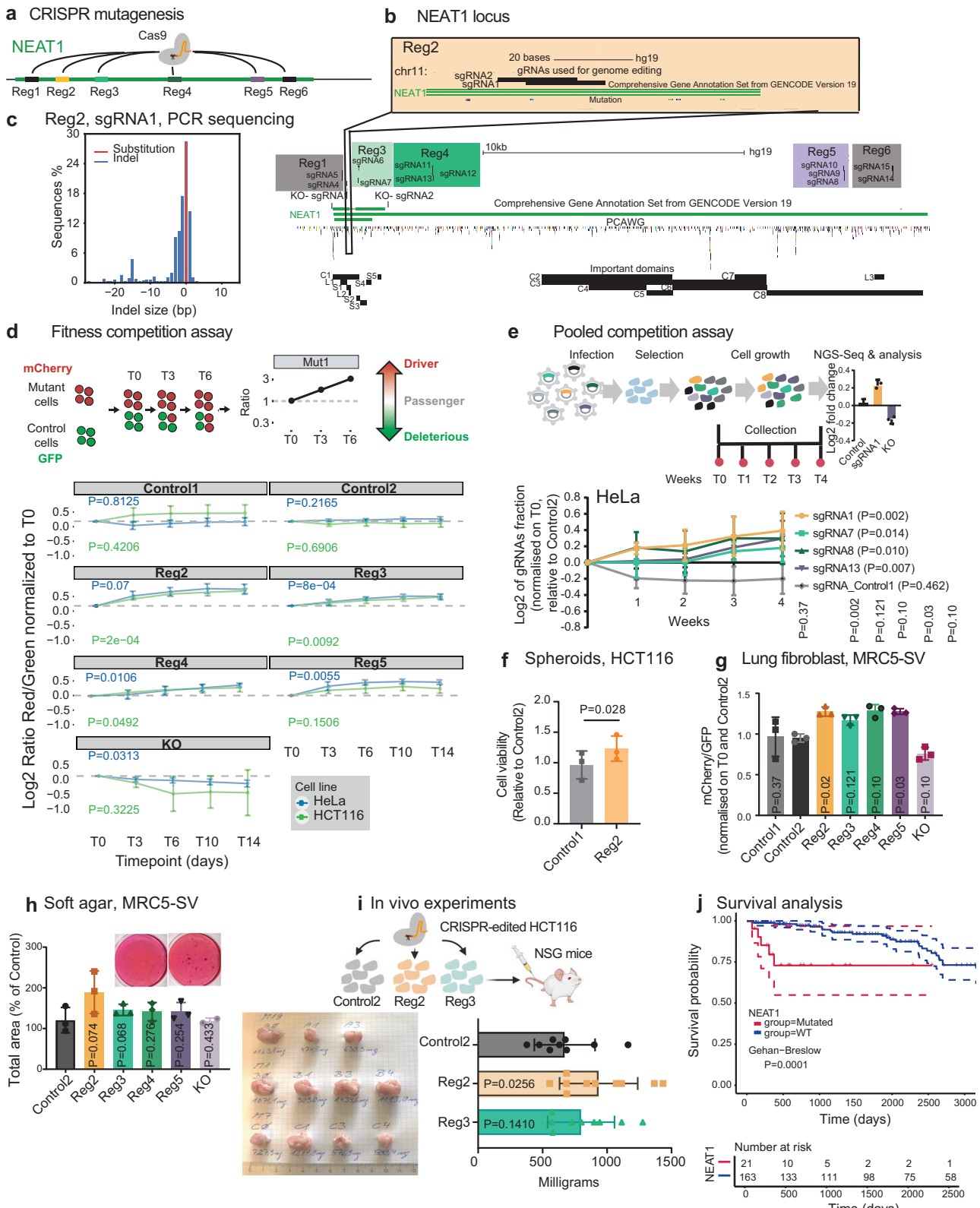

could interpret the functional relevance of highly mutated lncRNAs, underlining the importance of experimental results presented here.

The ability of ExInAtor2 to identify cancer lncRNAs was demonstrated by extensive functional studies, including for two lncRNAs, *LOHAN1* (head and neck cancer) and *LOLI1* (hepatocellular carcinoma). Not only are both capable of promoting cancer cell growth in their wild-type form, but interestingly in the case of *LOLI1*, this activity is enhanced by tumour mutations. These findings underline the usefulness of driver analysis in identifying cancer lncRNAs.

Among the candidate driver lncRNAs, we identified was the widely studied *NEAT1*. Previous tumour sequencing studies have noted the elevated density of SNVs at this locus, but generally attributed them to passenger mutational processes, possibly a consequence of unusually high transcription rate[2,33,35,62]. Here, we have provided experimental

**Fig. 5 | Mutations in *NEAT1* promote cell fitness and correlate with survival.**
**a** Experimental strategy to simulate tumour-like mutations in the *NEAT1* gene by Cas9 protein. **b** Detailed map of the six *NEAT1* target regions and 15 sgRNAs. Paired gRNAs used for the deletion of NEAT1_1 are indicated as KO- sgRNA1 and KO-sgRNA2. Previously described NEAT1 functional regions are indicated below[52]. **c** Analysis of mutations created by Cas9 recruitment. The frequency, size and nature of resulting DNA mutations are plotted. **d** Competition assay to evaluate fitness effects of mutations. Above: Rationale for the assay. Below: Red/green ratios for indicated mutations. "Control1/2" indicate sgRNAs targeting AAVS1 region. "KO" indicates paired sgRNAs designed to delete NEAT1_1. *N* = 4 experiments were performed, and statistical significance was estimated by linear regression model on log2 values. The mean value −/+ SD is plotted. Replicates were performed at different times (experimental replicates). **e** Upper panel: Set-up of mini CRISPR fitness screen. HeLa cells are infected with lentivirus-carrying mixtures of sgRNAs. The sgRNA sequences are amplified and sequenced at defined timepoints. Lower panel: Abundances of displayed sgRNAs, normalised to the Control2 negative control. Statistical significance was estimated by linear regression model; *n* = 4

(experimental replicates). The mean value with SD is represented. **f** HCT116 cells were cultured as spheroids and their population measured. Data show the mean value −/+ SD of *n* = 4 (experimental replicates). Statistical significance was estimated using Student's one-sided *t* test. **g** As for (**d**), but with non-transformed MRC5 lung fibroblast cells at timepoint Day 14. Statistical significance was estimated by one-sided Student's *t* test. Data show the mean value -/+ SD of *n* = 3 (experimental replicates). **h** MRC5 cells were seeded in soft agar, and the area of colonies was calculated. The mean and SD of *n* = 3 experiments is shown (experimental replicates). **i** *NEAT1* mutations in Reg2 enhances cell growth in NSG mice. HeLa cells were mutated and then implanted subcutaneously. Resulting tumour weight is shown at 4 weeks post-transplantation. Statistical significance was estimated by one-sided Student's *t* test based on *n* = 9 animals. Experiments were pooled from two groups of animals studied at different times. Data show the mean value -/+ SD. **j** The survival time of 184 lymphoid cancer patients from PCAWG is displayed. Patients were stratified according to whether they have ≥1 SNVs in the *NEAT1* gene. Two-sided Gehan Breslow rank test with confidence interval style set to dotted lines (95% CI). Source data are provided as a Source Data file.

evidence, via naturalistic in cellulo mutagenesis with CRISPR-Cas9, that *NEAT1* SNVs reproducibly give rise to increased cell proliferation, in a range of backgrounds including non-transformed cells and in living mice. Other observations are worthy of mention. Firstly, amongst fitness-altering *NEAT1* SNVs, we only observed those that increase growth, and none that decreased it. Secondly, not all tested regions of *NEAT1* could host fitness-altering mutations, and these were clustered at previously-mapped functional elements in mature RNA[50,52]. Altogether, these findings suggest that tumour SNVs at particular regions of *NEAT1* are phenotypically non-neutral and capable of increasing cell fitness by altering the function of encoded RNA. The notion that the NEAT1 gene represents a vulnerability to tumorigenesis is further supported by our demonstration that patients carrying mutations in the gene have worse prognosis, as well as published transposon insertional mutagenesis screens in mouse[31].

The relatively well-understood role of *NEAT1* in assembling ribonucleoprotein phase-separated paraspeckle organelles afforded important insights into SNVs' molecular mechanisms. Introduction of tumour mutations at the gene's 5′ end impacted protein binding, including a significant loss of interaction with the RNA Polymerase II complex mediated by known *NEAT1* interactor TAF15. Other known protein interactions are potentiated in mutated RNA, suggesting that changes in paraspeckles may be mediated by both gains and losses of protein interactions. The biological relevance of protein partners discovered here, including U2SURP and SREK1, is strongly supported by the impact on paraspeckles and cell viability resulting from their knockdown. The fact that *NEAT1* mutations gave rise to increased numbers and sizes of paraspeckle structures, suggests a model where SNVs alter the assembly of *NEAT1* ribonucleoprotein complexes, thereby promoting paraspeckle formation and hence cell growth. It will be interesting in future to understand how broadly this *NEAT1* model of altered protein interaction applies to other driver lncRNAs, and to what extent mutations act via alternative pathways such as alterations in RNA structure and folding, or interactions with other nucleic acids or biomolecules.

Future studies will have to address a number of gaps and questions raised here. Firstly, the availability of larger tumour cohorts will afford greater statistical power for driver-lncRNA detection. Larger cohorts will also enable us to identify driver lncRNAs in more focussed and meaningful sub-cohorts, for example tumours stratified by grade, therapy response, sporadic vs hereditary. Further gains may be made by incorporating more relevant estimates of functional impact into ExInAtor. We experimented with implementing FI estimates from changes to RNA structure, yet observed no significantly enriched lncRNAs, perhaps due to the low accuracy of available secondary structure prediction methods[63,64]. Future FI schemes incorporating improved structure prediction and protein/nucleic acid binding are

likely to yield improved driver predictions. On the other hand, driver prediction methods like ExInAtor2 may be susceptible to a variety of false-positive phenomena, including small open reading frames (sORFs) encoding micropeptides. However, this is unlikely to impact the driver lncRNAs presented here on account of aggressive filtering of input annotations and manually checking of all driver predictions using PhyloCSF predictions[65].

While we have provided functional experimental evidence for effects on cell phenotype arising from SNVs, it will be important to replicate this in better models, notably by introducing precise tumour mutations into cellular genomes (e.g., by recent Prime Editing method)[66,67], and testing their effects in faithful tumour models, such as mice or tumour organoids[68,69]. Finally, key mechanistic questions remain to be answered, such as the precise protein partners whose interaction is altered to result in paraspeckle changes[33,70,71].

In summary, we have presented experimental evidence that fitness-boosting somatic tumour mutations can act via changes in lncRNA function. We have sketched a mechanistic outline of how this process occurs via altered protein interaction and changes to membraneless organelles, in this case, paraspeckles. Our catalogue of candidate driver lncRNAs across thousands of primary and metastatic tumours provides a foundation for future elucidation of the extent and mechanism of driver lncRNAs.

## Methods
### ExInAtor2 algorithm
ExInAtor2 is composed of two separate modules for detection of positive selection: one for recurrence (RE), comparing the exonic mutation rate to that of the local background; another for functional impact (FI), comparing the estimated functional impact of mutations to background, both estimated in exons.

As an improvement to the first version of ExInAtor[33], the RE module compares the number of observed exonic mutations against a distribution of simulated exonic counts (Supplementary Fig. S1a), obtained by random repositioning of the variants the between the exonic and background regions while maintaining the same trinucleotide spectrum. Background region is defined for each gene as introns plus 10 kb up and downstream, after removing nucleotides overlapping exons from any other gene. Exonic and background regions can be further filtered to remove any additional "masked" regions defined by the user. In this manuscript, this functionality was used to mask low mappability regions and gap regions obtained from the UCSC Genome Browser (Supplementary Data 1).

The use of local background and controlling for trinucleotide content is intended to avoid known sources of false positives arising from covariates in mutational processes and mutational signatures, such as replication timing, gene expression, chromatin state, etc.[36].

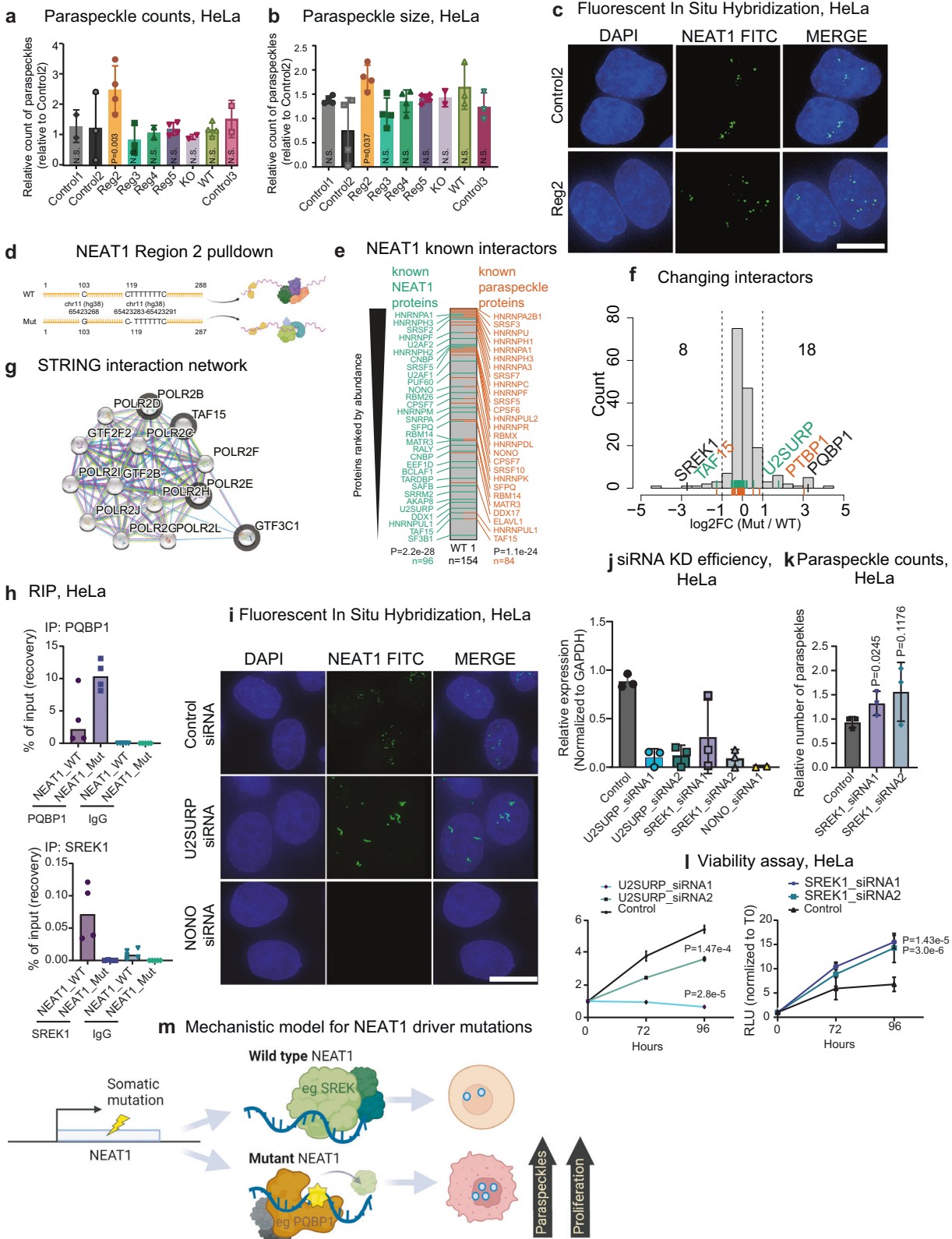

A *P* value is assigned to each gene, being the fraction of simulations with higher or equal number of mutations compared to the observed number (Formula 1).

$$RE_{p-value} = \frac{\#\,of\ simulated\ exonic\ counts \geq observed\ exonic\ count}{total\,\#\,of\ simulations} \quad (1)$$

*Formula 1*: *P* value calculation for the recurrence (RE) module.

The second FI module compares the mean functional score of the observed exonic mutations to a distribution of simulated values. Simulations are performed by random repositioning of mutations in exonic regions, while maintaining identical trinucleotide content (Supplementary Fig. S1b). Similar to the RE model, a *P* value is obtained by comparing the number of simulations with an exonic mean

**Fig. 6 | Mutations at the 5′ end of *NEAT1* increase paraspeckle formation and alter its protein interactome. a** Counts of paraspeckles and **b** paraspeckle size in HeLa cells treated with indicated sgRNAs. Values were obtained from 80 to 100 cells per replicate, with *n* = 5 (experimental replicates performed at different times). Statistical significance was estimated using one-tailed paired *t* test. Data show the mean value +/− SD. **c** Representative images from fluorescence in situ hybridisation (FISH) visualisation of *NEAT1* in HeLa cells expressing sgRNAs for Control2 and *NEAT1* Region 2. *n* > 3 biological replicates. Scale bar = 10 μM. **d** Sequences of biotinylated probes used for the mass-spectrometry analysis of NEAT1-interacting proteins. **e** Proteins detected by wild-type (WT) *NEAT1* probe, filtered for nuclear proteins only, are ranked by intensity and labelled when intersecting previously detected NEAT1-interacting proteins (green) and paraspeckle proteins (orange). Statistical significance was calculated by one-tailed hypergeometric test (to background of nuclear proteins *n* = 6758). **f** Histogram shows differential detection of proteins comparing mutated (Mut) and wild-type (WT) probes. Dotted lines indicate log2 fold-change cutoffs of −1/+1. **g** STRING interaction network based on a subset of the proteins lost upon

mutation (grey borders) interacting with the RNA polymerase II core complex. **h** Validations by RNA immunoprecipitation using antibodies for PQBP1 and SREK1. **i** FISH using *NEAT1* probes (green) in HeLa cells treated with indicated siRNAs. Cell nuclei in blue (DAPI). Values were obtained from 80 to 100 cells per replicate, with *n* = 3 replicates (experimental replicates). Scale bar = 10 μM. **j** qRT-PCR measurement of RNA levels in HeLa cells after transfection of siRNAs targeting *U2SURP and SREK1* genes. Data show the mean value +/− SD of *n* = 3 independent experiments. **k** Paraspeckles area in HeLa cells treated with two independent siRNAs targeting *SREK1*. Measurements from 80 to 100 cells per replicate. Statistical significance was estimated using one-sided paired *t* test. Data shows the mean value +/− SD of *n* = 3 replicates (experimental replicates). **l** Cell viability in siRNA-transfected cells was measured at indicated timepoints. Data shows the mean value +/− SD of *n* = 3 replicates (experimental replicates). Statistical significance was estimated by using towtailed *t* test. **m** Proposed model by which somatic mutations in *NEAT1* gene can alter protein interactome, increase paraspeckle numbers and boost cell proliferation. Source data are provided as a Source Data file.

functional score higher or equal to the observed value (Formula 2). This module work with any base-level scoring method. Given its previous successful use and integrative nature, we selected the Combined Annotation Dependent Depletion (CADD) scoring system[70].

$$FI_{p-value} = \frac{\#of\ simulated\ exonic\ means \geq observed\ exonic\ mean}{total\#of\ simulations} \quad (2)$$

*Formula 2*: *P* value calculation for the Functional Impact (FI) module.

In a final step, RE and FI *P* values are combined using the Fisher method (Formula 3).

$$T = -2 * \left[ ln\left(RE_{p-value}\right) + ln\left(FI_{p-value}\right) \right] \quad (3)$$

The Combined *P* value can be computed from the test statistic T which follows a chi-square distribution with degrees of freedom equal to 2n, where n is the number of tests being combined. Here in this case *n* = 2.

*Formula 3*: Fisher method for *P* value integration.

### Tumour somatic mutations

The principal source of mutations were primary tumours from the Pan-Cancer Analysis of Whole Genomes (PCAWG) project[1]. This dataset was created according to a uniform and strict methodology, including collection of samples, DNA sequencing and somatic variant calling, aggressive filtering to remove potential artefacts and false-positive mutations[1]. For practical reasons, we only considered Single Nucleotide Variants (SNVs) arising from substitutions, insertions and deletions of length 1 bp (indels) (Fig. 1b). After this filtering, the PCAWG dataset comprises 37 cancer cohorts, 2583 samples and 45,703,485 SNVs (Fig. 1b). Analyses were performed either on individual cohorts, or on the "Pancancer" union of all cohorts.

### Gene annotation and filtering

We employed a filtered lncRNA gene annotation based upon GENCODE annotation. Beginning with GENCODE v19 annotation, we discarded lncRNA genes overlapping protein-coding genes, or containing at least one transcript predicted to be protein-coding by CPAT[71], with default settings of coding potential >=0.364. To the remaining list of 6981 genes, we added 294 genes from Cancer LncRNA Census (CLC)[26], not annotated in GENCODE v19. The resulting set of 7275 lncRNA genes were used here unless otherwise specified (Fig. 1c and Supplementary Data 2).

## ExInAtor2 benchmarking against other driver discovery methods

We collected driver predictions from ten methods, in addition to the combined predictions generated by the PCAWG driver group (PCAWG combined, PCAWGc) that displayed best overall performance[2]. We only selected PCAWG methods that were run in both protein-coding and lncRNAs, and for which predictions were available for individual cohorts (Fig. 2a).

The original PCAWG publication used carefully filtered annotations for protein-coding and lncRNA genes[2]. Only coding sequences (CDS) of protein-coding genes were considered, while lncRNAs were strictly filtered by distance to protein-coding genes, transcript biotype, gene length, evolutionary conservation and RNA expression. For benchmarking, we ran ExInAtor2 using the same PCAWG annotations.

## Evaluation of *P* value distributions

Under the assumption that most genes are not cancer drivers and follow the null distribution, the collection of *P* values should mimic a uniform distribution with deviation of a small number of genes at very low *P* values[72]. Quantile–quantile plots (QQ-plot) (Fig. 2b and Supplementary Fig. S3a) display the observed and expected *P* values in −log10 scale. In order to generate the theoretical distribution for each driver method across all 37 cohorts and the Pancancer set, we ranked the total list of *n* observed *P* values from lowest to highest, then for each *i* observed *P* value we calculated an expected *P* value according to the uniform distribution (Formula 4).

$$expected_i = \frac{i}{n} \quad (4)$$

*Formula 4*: Expected *P* value calculation. *i* represents the rank of the corresponding observed p value in the total distribution of *n* observed *P* values, therefore *i* values range from 1 to *n*.

For each driver method, only genes with a reported *P* value were included in this analysis, i.e., NA cases were discarded. By visual inspection of the QQ plots, a correct observed distribution of *P* values should follow a line with 0 as intercept and 1 as slope, where extreme values beyond approximately 2 in the *x* axis should deviate above the diagonal line. We used the MLFC (Formula 5) to numerically estimate such deviation and evaluate the performance of driver gene predictions[72]. The closer to zero the MLFC, the better the statistical modelling of passenger genes following the null distribution[72].

$$MLFC = \frac{1}{n} * \sum_{i}^{n} |\left(\frac{observed_i}{expected_i}\right)| \quad (5)$$

*Formula 5*: MLFC. *n* represents the total number of *P* values an *i* the lowest *P* value.

## Gene benchmark sets

We downloaded known driver genes from the Cancer Gene Census[39] (CGC) (www.cancer.sanger.ac.uk/census) on 06/02/2019 as a TSV file. We extracted all GENCODE *ENSG* identifiers, resulting in a list of 703 genes. For lncRNAs, we used the second version of the Cancer LncRNA Census[26], which contains 513 GENCODE lncRNAs.

## Precision, sensitivity and F1 comparison

CGC and CLC genes were used as ground truth for driver predictions of protein-coding and lncRNAs, respectively. Three metrics were used to compare driver predictions: Precision, the proportion of predictions that are ground truth genes (Formula 6); Sensitivity, the fraction of ground truth genes that are correctly predicted (Formula 7); F1-score, the harmonic mean of precision and sensitivity (Formula 8).

$$Precision = \frac{TP}{TP + FP} * 100 \qquad (6)$$

*Formula 6*: Precision.

$$Sensitivity = \frac{TP}{TP + FN} * 100 \qquad (7)$$

*Formula 7*: Sensitivity.

$$F1 - score = 2 * \frac{Precision * Sensitivity}{Precision + Sensitivity} \qquad (8)$$

*Formula 8*: F1-score.

## Simulated mutation datasets

To generate realistic simulated data, each mutation was randomly repositioned to another position with an identical trinucleotide signature (ATA > ATA, being the central nucleotide the one mutated) within a window of 50 kb on the same chromosome.

## Generation and comparison of genomic features

Evolutionary conservation: We downloaded base-level PhastCons scores for all 46-way and 100-way alignments[73] from the UCSC Genome Browser[74]. We calculated the average value across all exons of each gene.

Expression in normal samples: We obtained RNA-seq expression estimates in transcripts per million (TPM) units for 53 tissues from GTEx (https://gtexportal.org/home/datasets). For tissue specificity, we calculated *tau* values as previously described[75] (https://github.com/severinEvo/gene_expression/blob/master/tau.R).

Replication timing: We collected replication time data of 16 different cell lines from the UCSC browser[74] (http://genome.ucsc.edu/cgi-bin/hgFileUi?db=hg19&g=wgEncodeUwRepliSeq).

miRNA binding: We downloaded both bioinformatically predicted (miTG scores) and experimentally validated miRNA binding to lncRNAs from LncBase[76] (http://carolina.imis.athena-innovation.gr/diana_tools/web/index.php?r=lncbasev2%2Findex).

Tumour expression: Expression values in units of FPKM-uq were obtained from PCAWG[1].

Drug-expression association: We extracted expression-drug association *P* values from LncMAP[77] (http://bio-bigdata.hrbmu.edu.cn/LncMAP).

Germline cancer small nucleotide polymorphisms (SNPs): We downloaded SNPs from the GWAS Catalogue[78] (https://www.ebi.ac.uk/gwas/).

CIS evidence in mice: We downloaded CIS coordinates from CCGD[79] (http://ccgd-starrlab.oit.umn.edu/download.php) and mapped them to human hg19 with LiftOver (https://genome.ucsc.edu/cgi-bin/hgLiftOver) from the UCSC browser[74]. Then, we calculated the number of CIS intersecting each lncRNA divided by the gene length with a custom script using BEDtools[80]. CIS per Mb values are available in Supplementary Data 3.

Additional information on driver lncRNAs is provided in Supplementary Data 6.

## Survival analysis

Survival plots were constructed using donor-centric whole-genome mutations dataset, overall survival data and tumour histology data from UCSC Xena Hub: https://xenabrowser.net/datapages/?cohort=PCAWG%20(donor%20centric)&removeHub=https%3A%2F%2Fxena.treehouse.gi.ucsc.edu%3A443. The whole-genome mutations file was intersected with comprehensive gene annotation v38liftv37 (https://www.gencodegenes.org/human/release_38lift37.html) using BEDtools intersect to isolate donors with mutations in lncRNA of interest. Survival of donors with mutations in lncRNA of interest was then compared against the group of donors without mutations in lncRNA of interest using R packages "survival" (https://cran.r-project.org/web/packages/survival/index.html) and "survminer" (https://cran.r-project.org/web/packages/survminer/index.html)[81]. Log-rank test was used to compare the survival times of the two groups. For lymphoid tumours, patients with 40000 mutations or more were stripped from the analysis.

## *NEAT1* structure and element analysis

**Elements.** The window spanning 300 bp around Mut1a and Mut1b (hg19 chr11:65190589-65190888; hg38 chr11:65423118-65423417) was annotated with the programme ezTracks[82] using the following datasets as input: (i) structural features: RNA structures conserved in vertebrates (CRS)[83], DNA:RNA triplex structures[84], R-Loops lifted over to hg38[85]; (ii) conservation: phastCons conserved elements in 7, 20, 30 and 100-way multiple alignments[73] retrieved from UCSC genome browser[86]; (iii) high confidence narrow peaks from eCLIP experiments from ENCODE[87] (Complete list of accessions is located at Supplementary Data 9.

**RBP motif mapping.** The 20 bp-padded sequence around Mut1a and Mut1b (hg19 chr11:65190719-65190775) was extracted and then used to generate the sequence of the three distinct alleles WT, only Mut1a and only Mut1b. The three sequences were used as input for de novo RBP motif matching in the web servers RBPmap[88] using the option Genome: other and all Human/Mouse motifs) and RBPDB[89] (using the default score threshold, 0.8). Outputs were manually parsed and further processed using an in-house Python script.

**SNP structural impact analysis.** Sequences for the window spanning 300 bp around each mutation target were extracted. Then, only substitutions were kept and encoded according to their relative position and submitted to the MutaRNA web server[90], which also reports scores from RNAsnp[63].

## Mutual exclusivity and co-occurrence

We used DISCOVER[91] to analyse mutations in long noncoding RNA and coding regions from publicly available PCAWG data. Coding mutations were obtained from the following source (consensus coding mutation calls): https://xenabrowser.net/datapages/?dataset=October_2016_whitelist_2583.snv_mnv_indel.maf.coding.xena&host=https%3A%2F%2Fpcawg.xenahubs.net&removeHub=https%3A%2F%2Fxena.treehouse.gi.ucsc.edu%3A443.

Noncoding mutations was obtained from the following source (whole-genome somatic mutation calls): https://xenabrowser.net/datapages/?dataset=October_2016_whitelist_2583.snv_mnv_indel.maf.xena.nonUS&host=https%3A%2F%2Fpcawg.xenahubs.net&removeHub=https%3A%2F%2Fxena.treehouse.gi.ucsc.edu%3A443.

Noncoding mutations were intersected with exonic regions of lncRNA genes (GENCODE v42lift37) and further filtered to include only SNVs (mutations of length 1).

Genes were recorded as mutated (1) or wild-type (0) across donors in PCAWG that had data available for both ($n = 1750$) noncoding ($n = 1782$) and coding mutations ($n = 2550$) (to avoid confounding the analysis by including donors with data unreleased to the public).

Mutual exclusivity and co-occurrence analysis was performed for 17 cancer-driver lncRNAs discovered from PCAWG data and 572 known protein-coding cancer-driver genes from COSMIC database (Tier-1)[92].

## Cell culture

HeLa, HEK293T and HCT116 were a kind gift from Roderic Guigo's lab (CRG, Barcelona). The MRC5-SV cells were provided by the group of Ronald Dijkmanthe (Institute of Virology and Immunology, University of Bern) and the HN5 tongue squamous cell carcinoma cells by Jeffrey E. Myers (MD Anderson) to Y. Zimmer. SNU-475 were purchased from ATCC (#crl-2236). HuH7 were purchased from Cell Line Service (#300156). All the cell lines were authenticated using Short Tandem Repeat (STR) profiling (Microsynth Cell Line Typing) and tested negative for mycoplasma contamination.

HeLa, HN5 and HEK 293 T cell lines were cultured at 37 °C in 5% $CO_2$ in Dulbecco's Modified Eagle's Medium high-glucose (Sigma) supplemented with 10% FBS (Gibco), 1% L-glutamine (Thermo Fisher), 100 IU/mL of Penicillin/Streptomycin (Thermo Fisher).

HCT-116 and MRC5-SV were cultured in McCoy (Sigma) and EMEM (Sigma), respectively, both supplemented with 10% FBS (Gibco), 1% L-glutamine (Thermo Fisher), 100 IU/mL of Penicillin/Streptomycin (Thermo Fisher). SNU-475 (ATCC) and HuH7 (Cell Line Service) hepatocellular carcinoma cell lines were cultured at 37 °C in 5% $CO_2$ in RPMI-1640, Glut Amax™ (Gibco) supplemented with 10% FBS (Gibco) and 100 IU/mL of penicillin/streptomycin (Thermo Fisher).

IHH was kindly generated and provided by Professor Didier Trono and Dr Tuan Nguyen[93] and were cultured in DMEM-F12 (Sigma) medium supplemented with 20 mU/ml insulin (Novo Nordisk), 50 nM Dexamethasone (Sigma) 100 IU/mL of penicillin/streptomycin (Thermo Fisher) and 10% FBS (Gibco).

## Primary human hepatocytes

Perfusable wedges of normal human liver tissue were obtained from the periphery of liver specimens from patients undergoing surgical resection. Human liver tissue was obtained from patients undergoing surgical resection for colorectal metastasis. Signed informed consent was obtained from all patients in accordance with institutional guidelines and according to study approval of the Ethics Commission of the Canton of Bern. Cells were isolated using a two-step enzymatic perfusion protocol as previously described[94]. The hepatocytes were seeded at a density of 90,000/cm² onto six-well tissue culture plates coated with rat tail collagen in Dulbecco's modified Eagle medium containing 10% foetal bovine serum (Life Technologies, Switzerland), left to attach for 2 h, then washed twice with phosphate-buffered saline (PBS, Life Technologies, Switzerland) to remove unattached cells. The hepatocytes were cultured in Williams-E, supplemented with insulin (0.015 IU/mL, NovaRapid, Novo Nordisk), hydrocortisone (5 µmol/L, Sigma-Aldrich, Buchs, Switzerland), penicillin, streptomycin, glutamine (100 IU/mL, 100 µg/mL, 2 mmol/L, GPS, Life Technologies, Switzerland) for 24 h before use.

Cultured hepatocytes were washed with PBS and transduced for 16 h with 5 µl of concentrated lentiviral vector in 2 ml complete Williams-E medium with a supplement of 50 µM vitamin E succinate (Sigma). 4 days after inoculation, transfection was confirmed with imaging of the GFP tag with Leica Stellaris 8 (Leica). RNA was extracted with NucleoZOL according to the manufacturer's instruction (Macherey–Nagel) and quantified by Nanodrop analysis. After reverse transcription with Omniscript reverse transcriptase (Qiagen), the

mRNA was quantified using TaqMan gene expression assay reagents (Sigma-Aldrich) and primers from Thermo Fisher. Human probes include CCNA2(Hs00996788_m1), CCNB1(Hs01030099_m1), CCNE1(Hs01026536_m1) and BIRC5(Hs04194392_s1). Eukaryotic 18 S (4352930E) was used as a normaliser. Amplifications were performed with QuantStudio 7 according to standard protocol (Thermo Fisher), and the relative changes in mRNA expression were calculated using the ΔΔCT method.

## Gene overexpression and knockdown experiments

Both the wild-type and mutated lncRNA spliced sequences were synthesised by Gene Universal Inc, into pcDNA3.1 vector backbone. Control pcDNA3.1 plasmids contained the sequence of enhanced green fluorescent protein (EGFP).

Transfection in HN5 cells: For each transfection, 1.6 µg of plasmid DNA has been incubated for 20 min with 4 µl of Lipofectamine 2000 transfection reagent (Invitrogen) in 0.2 ml of OptiMEM media (Gibco) and added to the cells cultured in a six-well plate. As all plasmids contain G418 resistance gene, cells were cultured in 2.5 mg/ml of G418 (Gibco) 48 h after transfection. After ~10 days, when the antibiotic selection was over, we collected cell pellets to extract the RNA and test the overexpression efficiency.

Transfection in HuH7 cells: For each transfection, 100 ng of plasmid DNA were incubated for 20 min with 0.15 µl Lipofectamine 3000 and 0.2 µl P3000 transfection reagent (Invitrogen) in 10 µl RPMI-1640, GlutaMAX™ (Gibco) and added on top of 2000 HuH7 cells cultured in a 96-well plate.

For both cell lines, after ~10 days, when the antibiotic selection was complete (as judged by 100% death of untransfected cells), we collected cell pellets to extract the RNA and test the overexpression efficiency.

Knockdown in SNU-475 and HuH7 cells: For the transfections, 10 nM of each ASO were incubated with 0.15 µl Lipofectamine 3000 (Invitrogen) for 20 min in 10 µl RPMI-1640, GlutaMAX™ (Gibco) and added on top of 2000 SNU-475 or HuH7 cells cultured in a 96-well plate. Transfection efficiency was measured with qPCR after 144 h.

ASO sequences are available in Supplementary Data 4.

## Crystal violet staining

Cells were dissociated with 0.05% trypsin-EDTA (Gibco), resuspended in complete media and counted in Neubauer chamber. Subsequently, 1000 cells per well were plated in a six-well plate, cultured for 1 week and stained in a 2% Crystal violet (Sigma) solution. The area percentage covered with cells was analysed using ImageJ (%Area). Data analysis was conducted in GraphPad Prism version 8.0.1. One-way ANOVA was used to determine statistical significance, alpha = 0.05.

## Proliferation assay−SNU-475 and HuH7

After transfection and neomycin selection, the proliferative capacity of SNU-475 and HuH7 was measured every 24 h by resazurin assay. Briefly, Resazurin sodium salt (Sigma) was added to each well to reach a final concentration of 3 µM and was incubated at 37 °C for 2 h. Absorbance was measured with Tecan Spark Plate Reader at 545 nm and 590 nm.

## CRISPR sgRNA design and cloning

CRISPR activation in HeLa cells was performed as described by Sanson and colleagues[95]. sgRNAs were designed using the GPP sgRNA Designer CRISPRa from the Broad Institute (https://portals.broadinstitute.org/gpp/public/) (Supplementary Data 4). For each sgRNA, forward and reverse DNA oligos were synthesised, introducing the BsmB1 overhangs. The two oligos were phosphorylated with the Anza™ T4 PNK Kit (Thermo Fisher) according to the manufacturer's instructions in a 10 µl final volume. The phosphorylation/annealing reaction was set up in a thermocycler at 20 °C for 15 min, followed by

95 °C for 5 min and then ramp down to 25 °C at 5 °C/min rate. For ligation of annealed oligos into the pXPR_502 backbone (Addgene #96923), the plasmid was first digested and dephosphorylated with FastDigest BsmBI and FastAP (Thermo Fisher) at 37 °C for 2 h. Ligation reaction was carried out with the Rapid DNA Ligation Kit (Thermo) according to the manufacturer's instructions.

sgRNAs targeting *NEAT1* were designed using the GPP sgRNA Designer CRISPRKo from the Broad Institute (https://portals.broadinstitute.org/gpp/public/) (Supplementary Data 4), and cloned into the pDECKO backbone (Addgene #78534) as described above.

## Lentivirus production

For lentivirus production, HEK293T cells ($2.5 \times 10^6$) were seeded in poly-L-lysine coated 100-mm culture dishes 24 h prior to transfection. Cells were then co-transfected in serum-free medium with 12.5 μg of the plasmid of interest (Lenti dCAS-VP64_Blast plasmid or sgRNA-containing pXPR_502 or pDECKO), 4 μg of the envelope-encoding plasmid pVSVg (Addgene 12260) and 7.5 μg of the packaging plasmid psPAX2 (Addgene 8454) with Lipofectamine 2000 (Thermo Fisher) according to the manufacturer instructions. After 4–6 h the medium was replaced with complete DMEM. Virus-containing supernatant was collected after 24, 48 and 72 h post-transfection. The three harvests were pooled and centrifuged at 3000 rpm for 15 min to remove cells and debris. The supernatant was collected, and for every four volumes, one volume of cold PEG-it Virus Precipitation Solution was added. The mix was refrigerated overnight at 4 °C and centrifuged at $1500 \times g$ for 30 min at 4 °C. The supernatant was discarded, and the sample was centrifuged at $1500 \times g$ for 5 min. The lentiviral pellet was suspended in cold, sterile PBS, aliquoted into cryogenic vials and stored at −70 °C.

## Lentiviral transduction

**CRISPRKo.** For the generation and transduction of Cas9-expressing cell lines, HeLa, HCT116 and MRC5-SV Cas9 were incubated for 24 h with culture medium containing concentrated viral preparation carrying pLentiCas9-T2A-BFP and 8 μg/ml Polybrene. Twenty-four hours post-infection, antibiotic selection was induced by supplementing the culturing medium with 4 μg/ml blasticidin (Thermo Fisher) for 5 days. Blasticidin-selected cells were subjected to three rounds of fluorescence-activated cell sorting (FACS) to isolate high BFP-expressing cells.

**CRISPRa.** For the generation and transduction of dCas9-expressing cell lines, HeLa cells were incubated for 24 h with culture medium containing concentrated viral preparation carrying pLenti dCas9-T2A-BFP-VP64 and 8 μg/ml Polybrene. Cells underwent FACS sorting to enrich for high BFP-expressing cells.

**sgRNAs.** pLentiCas9-T2A-BFP or dCas9-T2A-BFP-VP64 stable cell lines were seeded into six-well plates at $10^6$ cells per well and supplemented with sgRNAs pDECKO or pXPR_502 lentiviral preps, respectively, and spinfected in the presence of polybrene (2 μg/ml) for 95 min at 2000 rpm at 37 °C, followed by medium replacement. Twenty-four hours post-infection, antibiotic selection was induced by supplementing the culturing medium with 2 μg/ml puromycin (Thermo Fisher) for at least 3 days.

## RT-qPCR gene expression analysis

HeLa cells were lysed, and total RNA was extracted by using the Quick-RNA™ Miniprep Kit (Zymo Research). For each sample, RNA was retro-transcribed into cDNA by using the GoScript™ Reverse Transcription System (Promega) and the expression of the target gene was assessed through Real-Time PCR with the GoTaq® qPCR Master Mix. To this purpose, target-specific mostly intron-spanning primers (listed in Supplementary Data 4) were designed by using the online tool Primer 3 version 4.1.0.

## Cell viability assay

After puromycin selection, cells expressing controls and candidates' guides were collected and seeded in 96-well plates in at least three technical replicates for each timepoint (3000 cells per well). Proliferation assay was performed using the Cell-Titer Glo 2.0 (Promega) reagent according to the manufacturer's instructions. Luminescence was measured with the INFINITE 200 PRO series TECAN reader instrument. Timepoint 0 (T0) reading was performed 4–5 h after cell seeding.

## 1:1 competition assay

HeLa, HCT116 and MRC5-SV cells were infected with pDECKO lentiviruses expressing fluorescent proteins. Control plasmids containing sgRNAs targeting *AAVS1* expressed GFP protein (pgRNAs-AASV1-GFP +), while the sgRNAs targeting the different regions of *NEAT1* expressed mCherry. After infection, and seven days of puromycin (2 μg/ml) selection, GFP and mCherry cells were mixed 1:1 in a six-well plate (150,000 cells). Cell counts were analysed by LSR II SORP instrument (BD Biosciences) and analysed by FlowCore software.

## Pooled competition assay

Screen: HeLa cells stably expressing sgRNAs targeting *NEAT1* Reg2, Reg3, Reg4, Reg5 and KO, and HeLa cells stably expressing sgRNAs Control1 and Control2 were counted and mixed in the following ratio 10:10:10:10:25:25. At day 0, 2 M cells were collected, while 2 M were plated and passaged every 2–3 days. Cells were harvested at 7, 14, 21 and 28 days for gDNA extraction. The experiment was conducted in six biological replicates.

Genomic DNA preparation and sequencing: Genomic DNA (gDNA) was isolated using the Blood & Cell Culture DNA Mini (< 5e6 cells) Kits (Qiagen, cat. no. 13323) as per the manufacturer's instructions. The gDNA concentrations were quantified by Nanodrop. For PCR amplification, 1 μg of gDNA was amplified in a 200 μl reaction using Q5® High-Fidelity 2X Master Mix (NEB #M0491). PCR master mix (100 μl Q5, and 10 μl of Forward universal primer, and 10 μl of a uniquely barcoded P7 primer (both stock at 10 μM concentration). PCR cycling conditions: an initial 30 s at 98 °C; followed by 10 s at 98 °C, 30 s at 68 °C, 20 s at 72 °C, for 22 cycles; and a final 2 min extension at 72 °C. NGS primers are listed in Supplementary Data 4. PCR products were purified with Agencourt AMPure XP SPRI beads according to the manufacturer's instructions (Beckman Coulter, cat. no. A63880). Purified PCR products were quantified using the Qubit™ dsDNA HS Assay Kit (Thermo Fisher, cat. no. Q32854). Samples were sequenced on a HiSeq2000 (Illumina) with paired-end 150 bp reads. The raw sequencing reads from individual samples were analysed by using a custom shell script to count the number of reads containing each sgRNA. The sgRNA counts were then normalised over the T0 and Control2.

## Deep sequencing to determine indel spectrum

Genomic DNA was extracted using the Blood & Cell Culture DNA Mini (<5 M cells) Kits (Qiagen, cat. no. 13323) as per the manufacturer's instructions. To prepare samples for Illumina sequencing, a two-step PCR was performed to amplify the different regions of *NEAT1*. For each sample, we performed two separate 100 μl reactions (25 cycles each) with 250 ng of input gDNA using Q5 MASTER MIX (NEB #M0491) and the resulting products were pooled (PCR reaction: 30 s at 98 °C; followed by 10 s at 98 °C, 30 s at 68 °C, 20 s at 72 °C, for 22 cycles; and a final 2-min extension at 72 °C). PCR amplicons were purified using solid phase reversible immobilisation (SPRI) beads, run on a 1.5% agarose gel to verify size and purity, and quantified by Qubit Fluorometric Quantitation (Thermo Fisher Scientific). The resulting DNA was used for reamplification with primers containing Illumina adaptors using the Q5 master Mix. Illumina adaptors and index sequences were added to 100 ng of purified PCR amplicon (PCR reaction: 30 s at 98 °C; followed

by 10 s at 98 °C, 30 s at 68 °C, 20 s at 72 °C, for 8 cycles; and a final 2 min extension at 72 °C).

## RNA-FISH and immunofluorescence

The FISH protocol was carried as previously described[96,97]. Briefly, HeLa cells grown on coverslips were fixed using 4% paraformaldehyde and permeabilised by 70% ethanol overnight. For RNA-FISH, Stellaris® FISH Probes, targeting Human *NEAT1* Middle Segment, labelled with FAM dye (1:100, Biosearch Technologies) were used and the procedure was carried out according to the manufacturer's instructions. Cells nuclei were counterstained with 1:15,000 DAPI (4′,6-diamidino-2-phenylindole) at room temperature and then mounted onto slides by using the VectaShield (Vector Laboratories) mounting media. Fluorescence signals were imaged at 100× (UPLS Apo 100×/1.40) using the DeltaVision Elite Imaging System and Softworx software (GE Healthcare). Images were acquired as Z-stacks, subjected to deconvolution, and projected with maximum intensity. Images were processed using a custom CellProfiler pipeline to determine paraspeckle number and size.

## Soft agar assay

The soft agar colony formation assay was performed as previously described[98]. Briefly, the assay was carried out in 6-well plates coated with a bottom layer of 1% noble agar in 2X DMEM (Thermo Fisher) supplemented with: sodium bicarbonate, 10% FBS (Gibco), 1% L-glutamine (Thermo Fisher), 100 IU/ml of Penicillin/Streptomycin (Thermo Fisher). Then, 7000 cells were suspended in 2× DMEM and 0.6% noble agar. The suspension mixture was subsequently applied as the top agarose layer. A layer of growth medium was added over the upper layer of agar to prevent desiccation. The plates were incubated at 37 °C in 5% $CO_2$ for 3 weeks until colonies formed. After 20 days, the colonies were stained with 200 ml of MTT [(3-(4,5-dimethylthiazol-2-yl)−2,5-diphenyltetrazolium bromide), (5 mg/ml), Sigma] and incubated for 3 h at 37 °C. The numbers of colonies were counted using the analysis software ImageJ.

## 3D spheroid assay

HCT116 stably expressing Cas9-BFP and sgRNA-mCherry targeting *NEAT1* locus were FACS sorted to enrich the population BFP + / mCherry + . The cells were allowed to grow for 7 days, then detached, counted and seeded onto Corning® 96-well Flat Clear Bottom White (Corning, cat. no. 3610) in 20 µl domes of Matrigel® Matrix GFR, LDEV-free (Corning, cat. no. 356231) and McCoy (Sigma, cat. No. M9309) growth medium (1:1) with a density of 10,000 cells per dome in four technical replicates. Matrigel containing the cells was allowed to solidify for an hour in the incubator at 37 °C before adding 80 µl of McCoy growth media on top of the wells. The spheroids were allowed to grow in the incubator at 37 °C in a humid atmosphere with 5% $CO_2$. After 4 h the number of viable cells in the 3D cell culture was recorded as timepoint 0 (T0), CellTiter-Glo® 3D Cell Viability Assay (Promega, cat. no. G9682) was added to the wells, following the manufacturer's instructions for the reading with the Tecan Infinite® 200 Pro. After 1 week, the measurement was repeated.

## RNA pulldown and mass spectrometry

RNA pull-down analysis was performed as previously described[99]. Briefly, wild-type and mutant *NEAT1* RNA fragments were transcribed in vitro using HiScribe™ T7 High Yield RNA Synthesis Kit (NEB, #E2040S) and labelled with Biotin using Biotin RNA Labelling Mix (Roche, #11685597910) according to the manufacturer's instructions. Biotinylated RNA (10 pmol) was denatured for 10 min at 65 °C in RNA Structure Buffer (10 mM tris-HCl, 10 mM $MgCl_2$, and 100 mM $NH_4Cl$) and slowly cool down to 4 °C. Nuclear fractions were collected as described previously (Carlevaro-Fita J. et al., 2018)[100] and precleared for 30 min at 4 °C using Streptavidin Mag Sepharose® (Sigma, #GE28-

9857-99) and NT2 Buffer [50 mM tris-HCl (pH 7.4), 150 mM NaCl, 1 mM MgCl2, 0.05% NP-40,1 mM DTT, 20 mM EDTA, 400 mM vanadyl-ribonucleoside, RNase inhibitor (0.1 U/µl; Promega), and l× protease inhibitor cocktail (Sigma)]. The precleared nuclear lysates (2 mg) were incubated with purified biotinylated RNA in NT2 buffer along with Yeast tRNA (20 µg/ml; Thermo Fisher Scientific #AM7119) with gentle rotation for 1.5 h at 4 °C. Washed Streptavidin Magnetic Beads were added to each binding reaction and further incubated at 4 °C for 1 h to precipitate the RNA-protein complexes. Beads were washed briefly five times with NT2 Buffer, and the retrieved proteins were then subjected to mass-spectrometry analysis, performed by the Proteomics & Mass Spectrometry Core Facility (PMSCF) of the University of Bern, Switzerland, using MaxQuant software for protein identification and quantification.

## Mass-spectrometry data processing

Intensity Based Absolute Quantification (iBAQ) and label-free quantitation (LFQ) intensities from the MaxQuant output were used for quantitative within-sample comparisons and fold-enrichment between-sample comparisons, respectively. A protein was considered enriched/depleted in a sample condition if its intensity was at least twofold greater/lesser than in the reference condition (proteins not detected in one of the conditions are imputed with the lowest value for that sample by MaxQuant). In addition, the resulting lists of proteins were filtered for nuclear localisation[101] to exclude potential false positives. To calculate the significance of the overlap with known *NEAT1* binding proteins[102–104] and known paraspeckle proteins[49] a hypergeometric test was applied to the background of all nuclear proteins ($n = 6758$). STRING was used for interaction analysis (physical subnetwork, minimum interaction score = 0.4, max number of direct interactors = 10) and GO term enrichment analysis[105]. Visualisation was performed with R version 4.1.1 and BioRender.com. Full mass-spectrometry data may be found in Supplementary Data 5.

## RNA immunoprecipitation (RIP)

**Cloning**. Unique hybridisation area (5′-CCGAGCGTAGTCCGAGCGTA-3′) was added to the 3′ end of wild-type *NEAT1* fragment in pcDNA3.1 expression construct, and unique hybridisation area (5′-CGAC-GAACGGTCCGATACGT-3′) was added to the 3′ end of mutant *NEAT1* fragment in pcDNA3.1 expression construct by quick change mutagenesis (primers used for cloning are listed in Supplementary Data 4).

**Overexpression in HeLa cells**. For each transfection, 1 µg of plasmid DNA (500 ng of pcDNA3.1 wild-type *NEAT1* fragment with unique hybridisation area and 500 ng of pcDNA3.1 mutant *NEAT1* fragment with unique hybridisation area) was incubated for 20 min with 7 µl of Lipofectamine 2000 transfection reagent (Invitrogen) in 0.3 ml of OptiMEM media (Gibco) and added to the cells cultured in a six-well plate. As all plasmids contain the Hygromycin resistance gene, cells were cultured in 0.5 mg/ml of Hygromycin B (Thermo Scientific™) 48 h after transfection.

RIP was performed with RNA ChIP-IT® kit (Active Motif, #53024) according to the manufacturer's instructions. Optimal conditions for RNA/chromatin shearing were 8 pulses of 20 s at 40% power with 30 s of rest on ice in between, for cells from one 15-cm plate. Immunoprecipitation was performed overnight using 10 µg of sheared RNA/chromatin and 2 µg of antibody. Antibodies used for immunoprecipitation were anti-SREK1 (Sigma, HPA037674), anti-PQBP1 (Bethyl Laboratories, A302-802A) and Recombinant Rabbit IgG, monoclonal [EPR25A]−Isotype Control (Abcam, ab172730). RNA was purified with Nucleozol (Macherey–Nagel) according to the manufacturer's instructions. GlycoBlue (Thermo Fisher) was used as coprecipitant at the isopropanol precipitation step to help visualise the pallet. The amount of RNA was measured by NanoDrop (Thermo Fisher).

**RT-qPCR analysis.** RT-qPCR was performed with same amount of input and immunoprecipitated RNA sample in each replicate (i.e., 32 ng, 50 ng, 70 ng and 100 ng in different replicates). Immunoprecipitated RNA transcripts and input samples were reverse transcribed to cDNA and amplified using Promega GoTaq® 1-Step RT-qPCR System (Promega) according to the manufacturer's instructions. Data were presented as % of input (recovery) = AE^(Ct input−Ct sample) × 100%. Where AE is the amplification efficiency of each primer pair. AE was calculated from the standard curves of each primer pair as =10^(−1/slope). Standard curves were generated from 21 measurements for each primer pair and increasing (known) amount of input sample covering the range from 1.25 to 386.52 ng. List of primers used is listed in Supplementary Data 4 For the MUT and WT *NEAT1*, the same forward primer was used, and reverse primers were reverse complements to unique hybridisation areas of each construct. Data analysis was conducted in GraphPad Prism version 8.0.1. *t* test was used to determine statistical significance, alpha = 0.05.

### siRNA experiments
Pre-designed siRNAs were purchased from Sigma (Cat. #SASI_Hs02_00364192 SREK1, siRNA1; #SASI_Hs01_00057195 SREK1, siRNA2; #SASI_Hs02_00314626 U2SURP siRNA; #SIC001-1NMOL universal negative control; #SASI_Hs02_00343477 NONO siRNA.

One day before transfection, $5 \times 10^4$ HeLa cells were plated in a 12-well plate in 1 mL of growth medium (DMEM) without antibiotics such that they will be 30–50% confluent at the time of transfection. In total, 40 pmol of siRNA oligomer were diluted in 100 µl OptiMEM I Reduced Serum Medium without serum, and the transfection was carried on with Lipofectamine 2000 (Thermo Fisher #11668019), according to the manufacturer's instructions. Cells were incubated the at 37 °C in a $CO_2$ incubator for 24 h before starting the phenotypic assays and 72 h before testing the gene knockdown.

### Mouse experiments
All animal experiments were carried out in accordance with and under the approval of the local experimental animal committee of the Canton of Bern and performed according to Swiss laws for animal protection. Animal care was provided in accordance with the procedures outlined in the Guide for the Care and Use of Laboratory Animals. The maximum tumour size of $1 cm^3$ was permitted by IACUC. This limit was not exceeded.

NSG mice were purchased from Charles River Laboratories; 6- to 8-week-old male and female mice were housed under specific pathogen-free conditions in individually ventilated cages with food and water provided ad libitum and were regularly monitored for pathogens. All animals used in the experiments were age- and sex-matched. In total, $2.5 \times 10^6$ HCT116 cells that had previously been mutated as described above using sgRNAs targeting *NEAT1* Region 2, *NEAT1* Region 3 and negative control region, were resuspended in HBSS and mixed at ratio of 1:1 with Matrigel (Cat. # 356231; Corning, NY, USA) followed by subcutaneous injection into the flanks of the mice. Animals were monitored every day after tumour implantation and animal health was scored using an animal health score sheet assessing parameters such as appearance, behaviour, body condition score index (BCS) bodyweight loss, mouse grimace scale, tumour size and tumour appearance. Tumour size was calculated as follows: (length × width × height) × π/6. If these parameters were associated with a cumulative score ≥5, mice were euthanized. In the experiment presented in this manuscript, all mice were euthanized at day 21, because individual mice in the control groups were scored ≥5. Mice were euthanized in their home cages with $CO_2$ using a standard operating procedure implemented at the Central Animal Facilities of the University of Bern. Sex was not considered in this study because it was not relevant for our driver mutation study, since NEAT1 is not located on Chromosome X. Experiments were approved by the local experimental animal committee of the Canton of Bern and performed according to Swiss laws for animal protection. The tumour weight was measured at day 21, when the animals were sacrificed.

### Reporting summary
Further information on research design is available in the Nature Portfolio Reporting Summary linked to this article.

## Data availability
Somatic mutation data: (1) The publicly available mutation WGS somatic and germline variant calls, mutational signatures, subclonal reconstructions, transcript abundance, splice calls and other core data generated by the ICGC/TCGA Pan-cancer Analysis of Whole Genomes Consortium are available for download at https://dcc.icgc.org/releases/PCAWG. Additional information on accessing the data, including raw read files, can be found at https://docs.icgc.org/pcawg/data/. In accordance with the data access policies of the ICGC and TCGA projects, most molecular, clinical and specimen data are in an open tier which does not require access approval. To access potential identification information, such as germline alleles and underlying sequencing data, researchers will need to apply to the TCGA Data Access Committee (DAC) via dbGaP (https://dbgap.ncbi.nlm.nih.gov/aa/wga.cgi?page=login) for access to the TCGA portion of the dataset, and to the ICGC Data Access Compliance Office (DACO; http://icgc.org/daco) for the ICGC portion. In addition, to access somatic single-nucleotide variants derived from TCGA donors, researchers will also need to obtain dbGaP authorisation. (2) The publicly available HMF data can be requested at https://www.hartwigmedicalfoundation.nl/en/data/data-acces-request/Somatic mutation data and clinical data: (3) The Pan-cancer Analysis of Whole Genomes Consortium (PCAWG) publicly available data used in this study for survival analysis, mutual exclusivity and co-occurrence are available in the UCSC-Xenahub database accessible at: [https://xenabrowser.net/datapages/?cohort=PCAWG%20(donor%20centric)&removeHub=https%3A%2F%2Fxena.treehouse.gi.ucsc.edu%3A443][106]. (4) The Mass Spectrometry data generated in this study are available via ProteomeXchange with identifier PXD034007. (5) The next-generation amplicon sequencing data generated in this study are available through the National Center for Biotechnology Information (NCBI) Short Read Archive (SRA) under Project Accession Number PRJNA966897. These data relate to two experiments: Deep sequencing to determine indel spectrum in NEAT1 CRISPR mutagenesis (Fig. 5c) and pooled competition assay (Fig. 5e). The remaining data are available within the Article, as Supplementary Information, or Data file. Source data are provided with this paper.

## Code availability
The code is accessible at https://github.com/gold-lab/ExInAtor2.git. In addition, the code has been deposited to Zenodo[107] and is publicly available.

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

## Acknowledgements

The results shown here are based upon data generated by the TCGA, PCAWG and GTEx consortia. We thank Iñigo Martincorena (Sanger Institute) for generously providing certain data analysis scripts. We thank Federico Abascal (Sanger Institute) for generously providing cancer cell fraction data. We thank Alina Naveed (DBMR) for helpful discussions about NEAT1. We acknowledge Anne-Christine Uldry and Manfred Heller of the Mass Spectrometry and Proteomics Laboratory at the University of Bern (PMSCF) for assisting with all mass-spectrometry aspects. We thank Basak Ginsbourger (DBMR) for administrative support, and Willy Hofstetter and Patrick Furer (DBMR) for logistical support. All computation was performed on the Bern Interfaculty Bioinformatics Unit computing cluster maintained by Rémy Bruggmann and Pierre Berthier. This publication and the underlying study have been made possible partly on the basis of the data that the Hartwig Medical Foundation has made available. This work was funded by the Swiss National Science Foundation through the National Centre of Competence in Research (NCCR) "RNA & Disease" (51NF40-182880), project funding "The elements of long noncoding RNA function" (31003A_182337), Sinergia project "Regenerative strategies for heart disease via targeting the long noncoding transcriptome" (173738); by the Medical Faculty of the University and University Hospital of Bern; by the Helmut Horten Stiftung, Swiss Cancer Research Foundation (4534-08-2018); and by Science Foundation Ireland through Future Research Leaders award 18/FRL/6194. This research was also funded by Science Foundation Ireland under Grant number [18/CRT/6214] (to S.R.) and in part by the EU's Horizon 2020 research and innovation programme under the Marie Sklodowska-Curie grant H2020-MSCA-COFUND-2019-945385.

## Author contributions

R.E., A.L. and R.J. conceived and designed the experiment procedure and performed data analysis and interpretation. A.L., S.R. and K.S. developed the software and performed most of the bioinformatics analysis. T.U., G.B., A.M., and F.M. performed the RNA pull-down experiments and NEAT1 FISH experiments. B.M.M., L.M., C.W., S.S., T.P., A.M.T., M.T. and N.B. performed the CRISPR experiments. I.B. performed the experiments regarding LOLI1 gene. L.H. performed the experiments regarding LOHAN1&2 genes. A.V. provided the set of known cancer genes. H.G.-R., A.A. and D.M. contributed to the bioinformatics analysis. M.R. performed the in vivo experiments. E.Z. and S.Z. performed functional experiments on cancer cell lines. M.K.J., Mi.M., Ma.M., Y.Z., D.S., A.F., C.R. and A.F.O. provided key inputs and tools. R.E., A.L. and R.J. wrote the manuscript with input from all the authors. P.P.M. supervised aspects of bioinformatics analysis.

## Competing interests

The authors declare no competing interests.

## Additional information

[1]Department of Medical Oncology, Inselspital, Bern University Hospital, University of Bern, 3010 Bern, Switzerland. [2]Department for BioMedical Research, University of Bern, 3008 Bern, Switzerland. [3]Institute of Genetics and Biophysics "Adriano Buzzati-Traverso", CNR, 80131 Naples, Italy. [4]Graduate School of Cellular and Biomedical Sciences, University of Bern, 3012 Bern, Switzerland. [5]School of Biology and Environmental Science, University College Dublin, Dublin D04 V1W8, Ireland. [6]Conway Institute for Biomolecular and Biomedical Research, University College Dublin, Dublin D04 V1W8, Ireland. [7]The SFI Centre for Research Training in Genomics Data Science, Dublin, Ireland. [8]Department of Visceral Surgery and Medicine, Inselspital, Bern University Hospital, University of Bern, Bern, Switzerland. [9]Department of Urology, Inselspital, Bern University Hospital, Bern, Switzerland. [10]Department of Radiation Oncology,

Inselspital, Bern University Hospital and University of Bern, Bern, Switzerland. [11]School of Molecular Sciences, University of Western Australia, Crawley, WA, Australia. [12]School of Human Sciences, University of Western Australia, Crawley, WA, Australia. [13]GENYO, Centre for Genomics and Oncological Research, Pfizer/University of Granada/Andalusian Regional Government, Granada 18016, Spain. [14]Instituto de Investigación Biosanitaria, Granada 18014, Spain. [15]Department of Biochemistry and Molecular Biology I, University of Granada, Granada 18071, Spain. [16]These authors contributed equally: Roberta Esposito, Andrés Lanzós. ✉e-mail: roberta.esposito@unibe.ch; rory.johnson@ucd.ie

