## [Peer Review File · Nature Communications]

Tumour mutations in long noncoding RNAs enhance cell fitnessReviewers' Comments:

Reviewer #1:

Remarks to the Author:

R. Johnson et al - Tumour mutations in long noncoding RNAs enhance cell fitness

(and apparently available as a preprint:

<https://www.biorxiv.org/content/10.1101/2021.11.06.467555v1.full>)

Major remarks:

-Provide evidence of transfection efficiency in HUH7 cells.

- Overexpression in HN5 was done with test after 48 hours, but for HUH7 the time was 120 hours. Explain the difference in the experimental timeframe between these two cell lines.

- There is nothing new here in finding relevance of NEAT1 to paraspeckles using mutagenesis of SNV sites. NEAT1 is one of the highest-expressed genes in humans, is known from the literature to be involved in the majority of cancers, hence the mutagenesis-induced cell proliferation result is trivial - and perhaps serves to justify including one of the co-authors' and their teams (Fox et al) on this paper. The burden of proof is on the authors to explain why it is not a trivial result. I am not sure that NEAT1 was exclusively seen as "passenger" in all previous studies of its relevance to cancer and never highlighted as an oncogene.

- Sven Diederichs published "The hallmarks of cancer – an lncRNA point of view" in 2012. Again, it's hard to believe that in the past 10 years there has been no new work on "investigating the existence of "driver" lncRNAs" in cancer. I think the paper is victim to selective referencing and is missing out on relevant prior art proving oncogenic lncRNAs in humans.

- Can this "ExInAtoR" pipeline bridge the gap between biologists (without programming skills) and bioinformaticians, in order to be a more user-friendly computational pipeline? Or is it just another tool that requires a biology-based Principal Investigator to hire a full-time programmer-bioinformatician to implement? If the latter, please note that the field is already oversaturated with such tools. In the 3rd post-genomic decade, we need more experiments to go from genome to personalized drugs, not more bioinformatic pipelines that don't add more than incrementally to existing knowledge.

- Can the Combined Annotation Dependent Depletion (CADD) scoring system, which the authors selected, be applied for assessing the functionality of the mutation (FI), and be suitable to evaluate the final impact of the mutation at RNA level for functional noncoding RNAs including lncRNAs without prior relevance to cancer (as are most in this paper, though not NEAT1) and miRNAs or their host genes?

- Pan-cancer, and to some extent cancer-specific, lncRNAs are going to be so numerous in other datasets – while their experimental validation remains so uncertain and so expensive – that some kind of prioritization or scoring, using a real-world system that takes lncRNA biology into account, is going to be essential for the authors' approach to be useful to biologists.

- Does the pipeline account for RNA secondary structure, and/or RNA-protein interaction (in case of interaction with protein partners, known from the literature or from public RNA-protein interaction datasets or databases if any)?

- Does this pipeline take into account the local annotation-based positional sequence context of the SNVs? The importance of particular SNVs in different parts of the gene, in coding sequence (CDS) vs UTR (for protein-coding genes), in promoters vs splice sites, exons vs introns etc (for all, including lncRNA, genes) is different. Can the authors' tool incorporate these annotation-based features in its rankings?
- Is the approach capable of considering: the stage of cancer, hereditary vs nonhereditary form of a particular cancer (e.g. breast cancer has different etiology if hereditary vs sporadic), drug response profile, progression, and grade? Can it pinpoint lncRNAs that are relevant to these properties, as opposed to merely new global cancer drivers?
- Does the pipeline consider the potential epistatic effect of different SNVs, or any interaction between two or more loci?
(given that the common disease, common variant context of SNPs as well as SNVs implies that individually-small additive effects from multiple loci combine to impact the phenotype)
- Lines 120-121: "...for driver genes amongst both non-protein-coding genes (including lncRNAs) and...". How do the authors deal with the fact that certain functional lncRNAs could encode for small functional peptides and may in fact function principally through these encoded peptides, not directly as RNAs? Is this distinction relevant to how the approach searches for "driver genes?"
- One of my key concerns about this pipeline is that the input data is one-dimensional because only genomic data can be used as input. This is yet another addition to the already-oversaturated field of cancer discovery pipelines that look at the genome but ignore the transcriptome, the proteome, and the epigenome. There is no accounting here for transcriptomics data and epigenetic data, even though circulating-RNA profiling from blood is becoming more common and even though expression profiles (in patients or at least in public datasets e.g. GTEx, FANTOM) and epigenome data (from patients or ENCODE etc) can really help prioritize the hits and eliminate false positives.

Minor comments:

- 1) Line 58: provide a reference for the first sentence
- 2) Line 83: add a "to" before comprehensively map
- 3) Line 401: references are not in subscript
- 4) Figure 6j: typo in the word " network "
- 5) Line 191: specify the bona fide genes and show the experiments related to these genes
- 6) Supp Fig 5d: typo in the word " features "
- 7) Line 830: incorrectly tagged to Supp File 4
- 8) Line 830: consider changing the word "file" to "figure" to eliminate confusion
- 9) Line 756: Insert a tab to create a new paragraph for "Knockdown"

Reviewer #2:

Remarks to the Author:

Esposito and colleagues present an improved version of ExInAtoR2, a pipeline to discover long

noncoding RNA (lncRNA) mutations in cancers, across a vast cohort of primary and metastatic tumours. The authors identify 54 'driver' lncRNAs (i.e., predicted to favor tumorigenesis) linked to cancer for the first time. The significance of this cohort of lncRNAs is supported by different clinical and genomic evidence. The authors follow up briefly on two lncRNAs, MILC and MIHNC, to show in cultured cells that when the endogenous wild-type lncRNAs are overexpressed, and when the mutant variants are expressed, cells display tumorigenic traits. They then focus on NEAT1, a lncRNA found here as bearing driver SNVs (single nucleotide variants). The authors express tumor-associated NEAT1 mutations in HeLa cells and observed an increase in protumor traits in both transformed and normal backgrounds.

This is an interesting study that reports the utility of ExInAto2 as a platform to find novel mutations in lncRNAs driving oncogenesis. The pipeline and the conclusions will interest cancer and lncRNA biologists. However, to solidify the value of their pipeline and strengthen the notion that lncRNAs are implicated in carcinogenesis, the authors should address several concerns.

Comments

1. In Figure 6, the differences in NEAT1 paraspeckle number and size are not very apparent. Are the differences significant? (no significance indicators are shown on the graphs) Furthermore, in tumors in which the SNVs are detected, are the size and number of NEAT1 paraspeckles different?
2. In Figure 6g, the authors used a 288-nt fragment for pulldown. To verify the proteins differentially interacting with NEAT1 WT and SNVs, the authors need to study if (1) the predicted proteins bind the lncRNAs differentially by using orthogonal assays (e.g., RIP or CLIP), and (2) differential binding is observed in tumors expressing NEAT1 SNVs.
3. Stronger evidence is needed that the variants are truly tumor-driving. Mouse xenografts expressing the proposed tumor variants will go a long way to support the authors' pipeline and conclusions.
4. The authors should at least begin to address how NEAT1 WT and SNV protein partners affect paraspeckle size and number, as well as how paraspeckle phenotypes may influence tumorigenesis.

Reviewer #3:

Remarks to the Author:

The study entitled "Tumour mutations in long noncoding RNAs enhance cell fitness" catalogs somatic mutations in long non-coding RNAs (lncRNAs) to identify candidate lncRNAs as oncogenic drivers. Furthermore, the effect of two candidates is analyzed as well as the impact of NEAT1 mutations on its function is characterized in greater detail. This large and important study is well written and structured and the claims are largely supported by the data presented. Nonetheless, several issues need to be addressed prior to publication in Nature Communications. In summary, three additional experiments should be performed: the oncogenic potential should be assessed for additional candidates, individual mutations should be studied for MILC / MINHC / NEAT1 (with a rescue) and the differential interaction of protein interactors with mutated NEAT1 should be verified.

Major issues:

1) Functional characterization of candidate drivers and mutations

The authors claim that "Driver mutations identify oncogenic lncRNAs with therapeutic potential" (l. 245), which is not currently justified in this generality by the data presented:

1a) The authors analyze in total three lncRNAs (MILC, MIHNC, NEAT1) out of the 54 mentioned in the abstract - which is a very low number and should be expanded especially given the ease to perturb these transcripts in tumor cell lines and determine cellular phenotypes like proliferation.

1b) In fact, one would understand and expect from the abstract that all 54 lncRNAs ("...display oncogenic potential when experimentally expressed...") would have been tested for their oncogenic potential - which is certainly not necessary, but should be clarified in the abstract.

1c) The differences in the proliferation potential are moderate at best for MILC (figures 4d, S4d) - transformation assays in non-transformed cells that would be closer to the determination of oncogenicity have not been performed. Hence, their functional relevance can be claimed, but it remains unanswered whether these are oncogenic drivers.

1d) For the overexpression experiment, the authors combined four (MILC) or five different (MINHC) mutations in one transcript rather than testing them individually. Have these four or five mutations been found together in one allele of one patient? Otherwise, these would need to be tested individually to at least get one step closer to the situation in the patient and see whether individual mutations indeed have an impact.

1e) For a "therapeutic potential", much more data would be needed including e.g. a therapeutic window between non-transformed and tumor cells, which is not addressed in the study - and also not necessary. However, all claims regarding therapeutic potential should be removed from the manuscript.

2) NEAT1

The mutations in the gene for the lncRNA NEAT1 are characterized in greater detail in a model system harboring mutations endogenously introduced by the CRISPR/Cas9 approach.

2a) Mutations have been introduced by non-homologous end joining repair and thus randomly at the selected targeted sites rather than by homologous recombination or prime editing resulting in precision genome editing. This raises the question how well these mixtures of mutations mimic the mutations found in patients. Are these specific mutations also found to a similar extent in patients? Preferentially, precision genome editing for at least the most frequent mutations in NEAT1 would be desirable to analyze actual patient-derived mutations in clonal cells.

2b) The CRISPR mutagenesis approach may also lead to off-target effects which have not been controlled for. Although multiple different sgRNAs have been used, a rescue experiment (e.g. knockdown) would thus be desirable for at least one selected mutation.

2c) In the interaction analysis, a fragment of only 288 nt is found to specifically bind to 154 different proteins. Then, changing only two nucleotides leads to changes for 26 proteins. Additional experiments should corroborate these findings. For example, a RIP experiment could be performed for two interactors - one enriched and one depleted - with wild-type and mutant NEAT1 transcripts to verify their differential interaction.

2d) In figure 5h, significance levels should be indicated. The relevance of figure 6f remains unclear.

Minor issues:

3) Results table

The manuscript and the practical use of the results would strongly benefit from a (supplementary) table presenting the most important results for the 54 lncRNAs selected including their name, identifier, localization, most important mutations, references on their function etc., so that all readers can immediately check for the lncRNAs of their interest.

4) References

The references #39 and #49 seem to be incomplete or incorrectly formatted.

5) Typos

Several figure labels harbor a number of typos which should be corrected (predicions, survivalal, etwork, fetures, recturm,...). Also the manuscript text should be checked again.

Response to Reviewers' Comments

NCOMMS-22-11205A-Z

Tumour mutations in long noncoding RNAs enhance cell fitness

Esposito, Lanzós et al

Reviewer #1, expertise in lncRNA biology, computational (Remarks to the Author):

We sincerely thank the Reviewer for providing thoughtful feedback on our work. While we don't entirely share all the sentiments expressed, nonetheless the many insightful suggested changes have enabled us to substantially strengthen our work, for which we are very grateful.

Major remarks:

1) Provide evidence of transfection efficiency in HUH7 cells.

The transfection efficiency was estimated by counting the number of GFP positive cells, upon transfection of the control plasmid expressing the GFP. This number has been estimated to be between 20-25% of cells. However, our plasmids all carried a neomycin resistance cassette. Thus, all the experiments started after the cells underwent neomycin selection at a concentration that had previously been confirmed to yield 100% death in non-transfected cells. This has been clarified in the Methods.

2) Overexpression in HN5 was done with test after 48 hours, but for HUH7 the time was 120 hours. Explain the difference in the experimental timeframe between these two cell lines.

We apologise for the lack of clarity regarding this point. The cell pellets to check the RNA over-expression of MILC and MIHNC genes were collected at the timepoint 0 (T0), defined as the day in which the cells were plated for the proliferation assays. The timeframe from transfection and T0 was different in the two cell lines: 10 days after transfection in HUH7 and 15 days for HN5, according to the days needed to select the cells resistant to neomycin (estimated by death of non-transfected cells), the selection marker in the overexpression plasmid. We have now clarified this in the Methods section.

3) There is nothing new here in finding relevance of NEAT1 to paraspeckles using mutagenesis of SNV sites. NEAT1 is one of the highest-expressed genes in humans, is known from the literature to be involved in the majority of cancers, hence the mutagenesis-induced cell proliferation result is trivial - and perhaps serves to justify including one of the co-authors' and their teams (Fox et al) on this paper. The burden of proof is on the authors to explain why it is not a trivial result. I am not sure that NEAT1 was exclusively seen as "passenger" in all previous studies of its relevance to cancer and never highlighted as an oncogene.

This question arises out of misunderstandings of nomenclature, which we here clarify. "Oncogenes" refer to any gene that can transform cells (Weinberg <https://acsjournals.onlinelibrary.wiley.com/doi/pdfdirect/10.3322/canjclin.33.5.300>), by either expression or mutation. "Driver mutations" are those genetic mutations capable of increasing cell fitness. Therefore, although there is substantial overlap, not all oncogenes are found to carry driver mutations. Further, the demonstration that a gene may act as an oncogene by overexpression, is *not* sufficient to infer that it is a driver gene. These are widely and routinely used definitions in the cancer genomics field.

Thus, NEAT1 is widely accepted to be an oncogene because its over-expression promotes tumour phenotypes. However, the existence of driver mutations altering NEAT1 activity *has never been demonstrated and is novel*. Indeed, Rheinbay et al (Rheinbay et al Nature 2020) concluded that observed NEAT1 mutations are phenotypically neutral:

“Overall, our findings suggest that the indels in MALAT1, NEAT1, ALB and MIR122 are not driver events and are the result of a transcription-associated mutational process. The previously reported oncogenic effect of altered MALAT1 and NEAT1 expression^{27,28,29} may thus be unrelated to these mutations. ”

It is thus the demonstration that NEAT1 is both an oncogene *and* a driver gene, whose mutations increase cell fitness, which is a key advance in this work.

We have now clarified and extended these points in the Introduction:

“Driver genes represent a subset of more broadly defined ‘cancer genes’, the latter defined as those that functionally promote or oppose oncogenic cell states, regardless of mutational status 7.”

As for the role of Prof Archa Fox in this work: We initiated collaboration with Prof Fox following our initial discoveries of mutations in NEAT1, not *vice versa*. We’ll be happy to provide email records to back this up if the Reviewer wishes.

4) Sven Diederichs published “The hallmarks of cancer – an lncRNA point of view” in 2012. Again, it’s hard to believe that in the past 10 years there has been no new work on “investigating the existence of “driver” lncRNAs” in cancer. I think the paper is victim to selective referencing and is missing out on relevant prior art proving oncogenic lncRNAs in humans.

We refer again to the definitions of “driver gene” and “oncogene”, explained in the above response. The Diederichs paper deals with cancer-promoting oncogenic lncRNAs (typically, whose expression levels are increased in tumour cells), *not* with “driver” lncRNAs carrying somatic fitness-altering mutations.

Indeed, if the Reviewer could provide us with references for said prior art demonstrating somatic fitness-altering SNVs acting through lncRNAs, then we would be very grateful for it. Otherwise, we feel justified in claiming novelty for this present study.

5) Can this “ExInAator” pipeline bridge the gap between biologists (without programming skills) and bioinformaticians, in order to be a more user-friendly computational pipeline? Or is it just another tool that requires a biology-based Principal Investigator to hire a full-time programmer-bioinformatician to implement? If the latter, please note that the field is already oversaturated with such tools. In the 3rd post-genomic decade, we need more experiments to go from genome to personalized drugs, not more bioinformatic pipelines that don’t add more than incrementally to existing knowledge.

Thank you for your views on the cancer genomics field. To summarise, the Reviewer here is asserting that:

1. Bioinformatic tools are invalid unless accessible to biologists with no programming skills;
2. Additional bioinformatic tools are not useful in the development of personalised medicine;

These views are outside of the mainstream of the cancer genomics field, one of the most successful branches of biomedicine of the past decade. Specifically:

- 1) “Can this “ExInAator” pipeline bridge the gap between biologists (without programming skills) and bioinformaticians”. We certainly agree that bioinformatic software should be accessible to non-bioinformatician, *when there is a need for this*. However, this is not the case for ExInAator: there is little need for driver discovery methods to be run by non-bioinformaticians. Driver discovery analyses are performed infrequently (not on a daily basis), on large, complex and infrequently produced datasets of tumour genomes, created by large consortia and hosted on remotely-accessed scientific computing servers. These are *not* analyses that would be performed by computer-illiterate wet lab biologists. Nonetheless, ExInator2 is in fact, one of the most straightforward to use driver discovery methods. It performs comparably to the PCAWG consensus predictions that requires running 10 separate and sometimes difficult to run software packages and statistically integrating their predictions. ExInator2 is made available to the highest standards of openness and documentation (Github), and indeed is already being used by bioinformatician collaborators who have complemented us on its ease of use. In summary, ExInAator is a practical software pipeline that is accessible to a wide audience of computer-literate biologists and bioinformaticians.

We have clarified the intended use of ExInAator2 now in the Discussion:

“ExInAator2 is straightforward to run by bioinformaticians or Unix-literate biologists, outcompetes most present-day tools, and is freely available in Github.”

- 2) “The field is over saturated with these tools”. While we respect the Reviewer’s personal opinion, it is not consistent with the facts: If the field were saturated, then the appearance of new tools would slow down, and new tools would have little space for improvement over their predecessors. On the contrary, the recent PCAWG publication achieved improved performance by integrating 10 recent driver methods (Rheinbay et al Nature 2020), demonstrating that no single method is optimal. ExInator2 alone performs comparably with the integrated PCAWG predictor, again demonstrating practical improvement. Similarly, a very recent paper in *Nature Communications* (Bostrom et al) demonstrated new and useful driver discovery strategy. Given the fundamental importance of driver gene catalogues in directing future personalised therapeutics, driver gene discovery remains a critical and justified focus of continued development efforts. We have updated the Introduction to more clearly explain these points:

“Computational driver gene discovery tools continue to be refined, yielding catalogues of increasing accuracy that form the foundation of precision therapeutic development”

6) Can the Combined Annotation Dependent Depletion (CADD) scoring system, which the authors selected, be applied for assessing the functionality of the mutation (FI), and be suitable to evaluate the final impact of the mutation at RNA level for functional noncoding RNAs including lncRNAs without prior relevance to cancer (as are most in this paper, though not NEAT1) and miRNAs or their host genes?

Yes, indeed, we applied CADD and our functional impact module to all lncRNAs. CADD integrates several lines of functionality like: conservation, epigenetic modifications and transcription factor motif disruption (<https://academic.oup.com/nar/article/47/D1/D886/5146191>). CADD score are the FI source

of choice for driver discovery methods at the PCAWG consortium (eg DriverPower and OncodriveFML) to discover non-protein-coding driver elements including lncRNAs (Link).

We did not investigate microRNAs here, since they have been the focus of other expert groups. However, we would expect that CADD scores would also be useful for them.

We are presently investigating other scoring methods, apart from CADD, that could provide more nuanced readouts of lncRNA functionality, and hope to implement these in a future version of ExInAator. See Comment 8 below for some additional comments in this direction.

7) Pan-cancer, and to some extent cancer-specific, lncRNAs are going to be so numerous in other datasets – while their experimental validation remains so uncertain and so expensive – that some kind of prioritization or scoring, using a real-world system that takes lncRNA biology into account, is going to be essential for the authors' approach to be useful to biologists.

We agree entirely that in future, additional prioritisation methods are likely to become important for driver lncRNAs.

To address this, we have performed preliminary work to test the feasibility of a ranking approach. We used genomic features (cancer and non-cancer features such as expression in cancer cell lines, expression in normal cell lines, abundance of cancer and non-cancer SNPs, etc.) indicative of lncRNA function, to rank the 17 cancer-driver lncRNAs from the PCAWG cohort. We employed widely-used Robust Rank Aggregation method to generate a unique ranking based on these diverse features (Figures R1.1, R1.2).

We note with interest that the top ranked driver lncRNAs by this method are SNHG3 and NEAT1, both widely-studied oncogenes. Therefore, genomic and biological features (cancer and non-cancer) can be used to prioritize candidate lncRNAs for experiments using ranking algorithms such as RRA. We plan to investigate these methods further in upcoming work, where we expect to identify far larger sets of driver lncRNAs than in the present work.

Figure R1.1: Putative driver lncRNAs discovered in PCAWG data were ranked using the indicated features derived from LnCompare dataset (Carlevaro-Fita et al).

	Ensembl ID	Gene Name	Score
1	ENSG00000242125	SNHG3	4.24E-02
2	ENSG00000245532	NEAT1	1.06E-01
3	ENSG00000259001	RPPH1	1.18E-01
4	ENSG00000258779	RP11-140I24.1	5.20E-01
5	ENSG00000251151	HOXC-AS3	5.54E-01
6	ENSG00000248202	RP11-455B3.1	5.77E-01
7	ENSG00000270022	RNU12	8.26E-01
8	ENSG00000271643	RP11-10C24.3	8.26E-01
9	ENSG00000251191	LINC00589	1.00E+00
10	ENSG00000229401	RP1-290I10.7	1.00E+00
11	ENSG00000249988	RP11-669M16.1	1.00E+00
12	ENSG00000241219	RP11-572M11.1	1.00E+00
13	ENSG00000260390	RP11-575I8.1	1.00E+00
14	ENSG00000266977	CTC-459F4.5	1.00E+00
15	ENSG00000270117	AP000769.7	1.00E+00
16	ENSG00000260780	RP11-580I1.1	1.00E+00
17	ENSG00000261763	RP11-442N1.2	1.00E+00

Figure R1.2: Rankings and RRA score of PCAWG driver lncRNAs. Note that lower RRA scores indicate greater confidence.

8) Does the pipeline account for RNA secondary structure, and/or RNA-protein interaction (in case of interaction with protein partners, known from the literature or from public RNA-protein interaction datasets or databases if any)?

Thank you for this insightful comment. Indeed, thanks to its flexible structure, ExInAto2 can accommodate secondary structure and protein partners in its functional impact (FI) module. In earlier studies, we tested whether RNA secondary structure estimated by the widely-used algorithm RNAsnp (<https://pubmed.ncbi.nlm.nih.gov/23315997/>) are useful for predicting driver lncRNAs. These results are displayed as a quantile-quantile plot in Response Figure R1.3 below. This analysis shows that RNA folding, at least as predicted by the present state of the art, provides little information with regards to functional impact of SNVs in lncRNAs. We hypothesise that this is due to the poor predictive power of conventional RNA folding algorithms, as recently discussed by Vicens and Kieft.

Regarding protein binding - this is an excellent suggestion. We are planning to investigate this in recently-started work on new cancer cohorts. We plan to investigate methods of integrating protein binding data, as the Reviewer suggests, as a means of refining functional impact estimation. However, we are still working on how to optimally use protein-binding data for FI, therefore it falls outside of the present study.

On the other hand, the CADD scoring system we employ is a carefully generated score incorporating multiple different sources of evidence, including evolutionary conservation, which has been widely used in the driver discovery field.

We have updated the Discussion to reflect these points:

“We experimented with implementing FI estimates from changes to RNA structure, yet observed no significantly enriched lncRNAs, perhaps due to the low accuracy of available secondary structure prediction methods 67,68. Future FI schemes incorporating improved structure prediction and protein / nucleic acid binding are likely to yield improved driver predictions.”

Figure R1.3: Quantile-quantile plot of p-values from the Pancancer cohort of PCAWG (other specific cohorts are similar) with RNAsnp scores in ExInAator2's functional impact module (not combined with recurrence module).

9) Does this pipeline take into account the local annotation-based positional sequence context of the SNVs? The importance of particular SNVs in different parts of the gene, in coding sequence (CDS) vs UTR (for protein-coding genes), in promoters vs splice sites, exons vs introns etc (for all, including lncRNA, genes) is different. Can the authors' tool incorporate these annotation-based features in its rankings? you

Thank for these interesting suggestions, which we address separately:

CDS / UTR: Since lncRNAs are by definition non-protein-coding, they have neither CDSs nor UTRs, therefore these concepts do not apply.

Splice sites: On the other hand, splice site mutations were an important subject of previous driver mutation searches. However, previous PCAWG work and our own unpublished work, found that splice site mutations are uninformative for driver discovery (<https://www.nature.com/articles/s41586-020-1965-x>), possibly due to their far smaller extent and hence statistical power. Therefore, we have not employed it here.

Promoters: The PCAWG consortium already extensively analysed protein-coding and lncRNA promoters in this context (Rheinbay et al). Rather than regulatory mutations, our work is focussed on mutations that impact mature RNA sequences, therefore we chose to focus on this aspect.

Exon / intron: This is precisely the rationale for how ExInAator2 works, and is the basis for its successful discovery of driver lncRNAs via exonic mutational load compared to background estimated from local flanking and intronic regions (Figure R1.4).

Figure R1.4: An example of genomic regions analysed by ExInAator2. The black regions are lncRNA exons, and the intronic / intergenic regions used to estimate background mutation rates are shown in grey. The example shown is a cancer-driver lncRNA hit (ENSG00000260390) predicted by ExInAator in PCAWG cohort.

10) Is the approach capable of considering: the stage of cancer, hereditary vs nonhereditary form of a particular cancer (e.g. breast cancer has different etiology if hereditary vs sporadic), drug response profile, progression, and grade? Can it pinpoint lncRNAs that are relevant to these properties, as opposed to merely new global cancer drivers?

Thank you for these excellent suggestions, several of which we are presently exploring for future follow up work. The ExInAator2 workflow is entirely compatible with the proposed analyses. Rather the bottleneck involves access to appropriate datasets: the availability of indicated clinical metadata, and sufficient number of patients (for statistical power).

Unfortunately, the relatively small size of the PCAWG cohorts make this type of analysis impractical in the present context.

Nonetheless to gain insights into this question, we investigated the possibility of stratifying patients by disease stage using a particularly large cohort available in PCAWG: Liver-HCC with 205 donors (filtered for hypermutated tumours and donors without any information on stage of Liver-HCC). We ran ExInAator on cohorts stratified by stages of Liver-HCC (1,2,3,4, All stages). In the subgroups, Stage 2 has the largest number of donors (100), followed by Stage 3 (55), Stage 2 (33) and Stage 4 (17). We discovered hits only in the largest subgroup, that is, Stage 2 of Liver-HCC (Figure R1.5) and all Stages combined (1+2+3+4). Four hits were discovered in Stage 2 of Liver-HCC of which two (Figure R1.6, R7; ENSG00000245532, ENSG00000263272) were discovered in the combined cohort (Figure R1.7), and two hits were specific to Stage 2 (Figure R1.6, ENSG00000184991, ENSG00000224739). It is intriguing to note that Stage 2-specific analysis identifies two candidates that are missed in the All Stages cohort, raising the possibility of stage-specific phenomena. These data indicate that ExInAator2 is indeed capable of identifying subtype-specific driver lncRNAs and this will be a promising avenue for future studies with larger cohorts and better-annotated clinical metadata.

We have updated the Discussion to mention these points:

“Larger cohorts will also enable us to identify driver lncRNAs in more focussed and meaningful cohorts, for example tumours stratified by grade, therapy response, sporadic vs hereditary.”

Figure R1.5: Stage-specific analysis of the hepatocellular carcinoma (HCC) cohort by ExInAto2.

Figure R1.6: Quantile-Quantile plot for putative cancer-driver lncRNAs in Stage 2 of Liver-HCC

11) Does the pipeline consider the potential epistatic effect of different SNVs, or any interaction between two or more loci? (given that the common disease, common variant context of SNPs as well as SNVs implies that individually-small additive effects from multiple loci combine to impact the phenotype)

Thank you for this excellent suggestion. The question of whether epistatic effects (co-occurrence or mutual exclusivity) can be found between lncRNA mutations is a fascinating one, which could yield mechanistic insights. The sensitivity of such analyses depends again on having sufficient data.

To address this, we have performed a thorough analysis of genetic interactions amongst mutations in both lncRNAs and protein-coding genes (PCGs). We used DISCOVER (Canisius et al) together with PCAWG data to analyse co-occurrence and mutual-exclusivity in lncRNA SNVs and known driver mutations in PCGs.

This analysis successfully identified known epistatic links between PCGs. However, we found no significant lncRNA-lncRNA or lncRNA-PCG epistatic interactions. These findings may point to a lack of epistasis, or simply limited statistical power given the size of the PCAWG cohort. These possibilities may be distinguished using future, larger datasets.

These findings represent the first, to our knowledge, analysis of lncRNA mutational epistasis, and are worthy of including in the publication. Thus we include this data in the new Supplementary Files 7 and 8, and mentioned in the Results:

“We also searched for evidence of epistatic interactions between SNVs in lncRNA drivers and other lncRNA or known protein-coding drivers. Although we could retrieve many known PCG-PCG interactions, both positive and negative, we found no example of an lncRNA SNV participating in such an interaction (Supplementary Files 7,8).”

12) Lines 120-121: “...for driver genes amongst both non-protein-coding genes (including lncRNAs) and...”. How do the authors deal with the fact that certain functional lncRNAs could encode for small functional peptides and may in fact function principally through these encoded peptides, not directly as RNAs? Is this distinction relevant to how the approach searches for “driver genes?”

It is true that some small peptides have been identified in lncRNAs, and could in principle yield false positive hits. However, this is not likely to significantly affect our study because:

1. The prediction of sORFs in lncRNAs is subject to high false positive rates that are often not correctly accounted for. In our own recent work (Lagarde et al), we only found evidence for small numbers of potential sORFs in lncRNAs;
2. In the present study, we stringently filtered out potentially protein-coding lncRNAs (see Methods);
3. Many of our hits display elevated mutations throughout their exons, and not just in one localised region as would be expected for a sORF;
4. We manually inspected all lncRNA hits for evidence of this and other artefacts, but could find none.

We have updated the Discussion accordingly:

“On the other hand, driver prediction methods like ExInAtoR2 may be susceptible to a variety of false-positive phenomena, including small open reading frames (sORFs) encoding micropeptides. However, this is unlikely to impact the driver-lncRNAs presented here on account of aggressive filtering of input annotations and manually checking of all driver predictions using PhyloCSF predictions”

13) One of my key concerns about this pipeline is that the input data is one-dimensional because only genomic data can be used as input. This is yet another addition to the already-oversaturated field of cancer discovery pipelines that look at the genome but ignore the transcriptome, the proteome, and the epigenome. There is no accounting here for transcriptomics data and epigenetic data, even though circulating-RNA profiling from blood is becoming more common and even though expression profiles (in patients or at least in public datasets e.g. GTEX, FANTOM) and epigenome data (from patients or ENCODE etc) can really help prioritize the hits and eliminate false positives.

We entirely agree that ultimately, cancer genomics can generate the best insights as an integrative discipline uniting diverse data, and this is a key feature of our study. In Figures 3d and 3e, we integrate multiple other ‘omics data in relation to these candidates: expression profiles (GTEx and TCGA), clinical profiles (survival and drug associations), and other genomics features (miRNA binding, GWAS SNPs). These features lend important

independent weight to the value of our driver lncRNA predictions.

Minor

comments:

1) Line 58: provide a reference for the first sentence

Done (<https://www.nature.com/articles/s41568-020-0290-x>)

2) Line 83: add a “to” before comprehensively map

Done

3) Line 401: references are not in subscript

Done

4) Figure 6j: typo in the word “ network “

Done

5) Line 191: specify the bona fide genes and show the experiments related to these genes

We have clarified the definition and source of these genes:

“defined as those validated by functional assays in vitro and in vivo from the scientific literature”

6) Supp Fig 5d: typo in the word “ features “

Done

7) Line 830: incorrectly tagged to Supp File 4

The indicated primers are listed in Supplementary File 4, tab “Oligos”.

8) Line 830: consider changing the word “file” to “figure” to eliminate confusion

The object in question is an Excel file containing all primers. We feel it is more appropriate to label it as a “file” rather than “figure”.

9) Line 756: Insert a tab to create a new paragraph for “Knockdown”

Done

Reviewer #2, expertise in lncRNA biology, experimental (Remarks to the Author):

Esposito and colleagues present an improved version of ExInAtor2, a pipeline to discover long noncoding RNA (lncRNA) mutations in cancers, across a vast cohort of primary and metastatic tumours. The authors identify 54 'driver' lncRNAs (i.e., predicted to favor tumorigenesis) linked to cancer for the first time. The significance of this cohort of lncRNAs is supported by different clinical and genomic evidence. The authors follow up briefly on two lncRNAs, MILC and MIHNC, to show in cultured cells that when the endogenous wild-type lncRNAs are overexpressed, and when the mutant variants are expressed, cells display tumorigenic traits. They then focus on NEAT1, a lncRNA found here as bearing driver SNVs (single nucleotide variants). The authors express tumor-associated NEAT1 mutations in HeLa cells and observed an increase in protumor traits in both transformed and normal backgrounds.

This is an interesting study that reports the utility of ExInAtor2 as a platform to find novel mutations in lncRNAs driving oncogenesis. The pipeline and the conclusions will interest cancer and lncRNA biologists. However, to solidify the value of their pipeline and strengthen the notion that lncRNAs are implicated in carcinogenesis, the authors should address several concerns.

We thank the Reviewer for thoughtful and useful suggestions for improvement.

Comments

1. In Figure 6, the differences in NEAT1 paraspeckle number and size are not very apparent. Are the differences significant? (no significance indicators are shown on the graphs) Furthermore, in tumors in which the SNVs are detected, are the size and number of NEAT1 paraspeckles different?

1) *Are the differences significant?* We have clarified the statistical significance in Figure 6. Only the NEAT1 Region 2 (Reg2) shows a significant increase in both number and size of paraspeckles ($P=0.003$ and $P=0.037$, 1-sided t -test). Indeed, the differences are reproducible across multiple independent experiments and are statistically significant. None of the other tested regions show a significant change in terms of paraspeckles. We have updated the figure with significance values.

2) *...in tumors in which the SNVs are detected, are the size and number of NEAT1 paraspeckles different?* This is a fascinating question. Unfortunately, we could find no dataset for which matched paraspeckle and mutations are available, therefore it is presently impossible to address this question.

2. In Figure 6g, the authors used a 288-nt fragment for pulldown. To verify the proteins differentially interacting with NEAT1 WT and SNVs, the authors need to study if (1) the predicted proteins bind the lncRNAs differentially by using orthogonal assays (e.g., RIP or CLIP), and (2) differential binding is observed in tumors expressing NEAT1 SNVs.

We agree it is important to independently validate the differential protein binding to WT and mutated NEAT1. We address the two points separately.

1) We agree that this is an important control and have performed the suggested

experiments. Using antibodies for one mutant-enriched (PQBP1) and one mutant-depleted protein (SREK1), we performed replicated RNA immunoprecipitation (RIP) experiments with wild-type (WT) and mutant (MUT) NEAT1 RNA (Figure R2.1). Firstly, we observe significant binding of NEAT1 to both proteins, compared to nonspecific IgG, as expected. Secondly, we see mutation-associated changes that agree with the mass spectrometry for both proteins: PQBP1 is more strongly associated with MUT NEAT1, while SREK1 more with WT NEAT1. These data have been included as a new Figure 6h in the manuscript.

Figure R2.1: RNA immunoprecipitation of NEAT1-bound proteins. Cells were transfected with vectors expressing wild-type (WT) or mutated (MUT) NEAT1 sequences. Immunoprecipitations were performed with antibodies against the indicated proteins. Control IgG antibody was used to estimate background.

2) This is an excellent suggestion. Unfortunately, we could find no tumour cohort dataset for which both mutations and protein binding are available. Therefore, it is impossible to address this at present.

3. Stronger evidence is needed that the variants are truly tumor-driving. Mouse xenografts expressing the proposed tumor variants will go a long way to support the authors' pipeline and conclusions.

Thank you for this excellent suggestion. We are happy to report that we have now performed indicated mouse xenograft experiments. Subcutaneous tumours, seeded by CRISPR-mutated HeLa cells, were grown in nude mice. Prior to injection, cells were mutated with either control sgRNA (Control2) or sgRNA targeting Region 2 (Reg2, sgRNA1) and Region 3 (Reg3, sgRNA7) of NEAT1. Using tumour weight at 21 days as an endpoint, we observed a statistically-significant increase in tumour mass arising from NEAT1 mutations in Reg2 ($P=0.0256$, one-sided Student's t -test). Reg3 displayed an increase that did not reach statistical significance ($P=0.142$). The data is displayed below (Figure R2.2) and now in the updated main figures (Figure 5i, Supplementary Figure S5e).

In summary, these results indicate that mutations in NEAT1 promote tumour cell fitness in a realistic *in vivo* setting. We believe that this first of its kind demonstration represents an important milestone and thank the Reviewer for suggesting this experiment.

4. The authors should at least begin to address how NEAT1 WT and SNV protein partners affect paraspeckle size and number, as well as how paraspeckle phenotypes may influence tumorigenesis.

Thank you for this suggestion. The broader question of how paraspeckles influence tumorigenesis is an emerging topic with the recent key publication establishing a genetic link between NEAT1 paraspeckles and tumorigenesis (Adriaens et al). Moreover, review articles have been written on the subject (for example, Link).

To gain further insights, we have performed new work to elucidate how NEAT1-protein interactions contribute to paraspeckle number and form. We selected differentially-bound proteins that had not been previously implicated in paraspeckle assembly: U2SURP and PQBP1 proteins (observed to preferentially bind MUT NEAT1) and SREK1 (observed to preferentially bind WT NEAT1). We used siRNA to knock down these proteins and evaluate the effect on paraspeckle formation and cell viability. NONO, known to be required for paraspeckle formation, was used as a positive control and its knockdown resulted in complete loss of paraspeckles (Figure R2.3).

We observed several striking effects on paraspeckles arising from knockdown of NEAT1 interactors. In the case of U2SURP, we observed the appearance of aberrant paraspeckle structures (Figure R2.3). A similar phenomenon has previously been observed by Ling Ling Chen's group for genes involved in mitochondrial functions and essential regulators of paraspeckles formation (<https://www.nature.com/articles/s41556-018-0204-2>). Interestingly, we did not observe any effect on cell proliferation, in line with results of NONO KD (Figure R2.4).

j Fluorescent In Situ Hybridization, HeLa

Figure R2.3: Fluorescence in situ hybridisation using NEAT1 probes (green) in HeLa cells treated with indicated siRNAs. Cell nuclei are stained with DAPI (blue).

Figure R2.4: Left: siRNA KD efficiency for the genes U2SURP, SREK1 and NONO. Right: Viability in HeLa cells, after knockdown of U2SURP and NONO genes, compared to control siRNA.

In the case of SREK1 loss of function, we observed a gain in paraspeckle numbers and accompanying increase in cell viability (Figure R2.5). For PQBP1 knockdown, we observed no detectable effect on paraspeckles or cell viability (not shown).

**j Paraspeckles counts,
HeLa**

**k Viability assay,
HeLa**

Figure R2.5: Effects of SREK1 loss of function by siRNA. Upper panel: paraspeckle number; Lower panel: cell viability.

Together, these findings support the relevance of our newly-discovered NEAT1-interacting proteins in paraspeckle formation, and suggest that mutation-induced changes to these interactions can feed forward into paraspeckle population and structure, and hence cell viability. We have developed a tentative model based on these findings, displayed as the new Figure 6m. The new data may be found in updated Figure 6i-l, Supplementary Figure S5k. We have updated the Results accordingly:

“Altogether, these findings suggest a model where tumour SNVs alter the protein-interactome of NEAT1, leading to both gains and losses of protein partners. For example, SREK1 appears to bind wild-type NEAT1 to repress formation of paraspeckles, and this interaction is abrogated by Region 2 mutations to boost paraspeckles and consequently accelerate cell proliferation (Figure 6m). It is likely that the gain and loss of other protein partners also contribute to mutation-associated changes in paraspeckle numbers and form.”

Reviewer #3, expertise in lncRNA biology, experimental and mass-spectrometry (Remarks to the Author):

The study entitled "Tumour mutations in long noncoding RNAs enhance cell fitness" catalogs somatic mutations in long non-coding RNAs (lncRNAs) to identify candidate lncRNAs as oncogenic drivers. Furthermore, the effect of two candidates is analyzed as well as the impact of NEAT1 mutations on its function is characterized in greater detail. This large and important study is well written and structured and the claims are largely supported by the data presented. Nonetheless, several issues need to be addressed prior to publication in Nature Communications. In summary, three additional experiments should be performed: the oncogenic potential should be assessed for additional candidates, individual mutations should be studied for MILC / MINHC / NEAT1 (with a rescue) and the differential interaction of protein interactors with mutated NEAT1 should be verified.

We wish to thank the Reviewer for these thoughtful and insightful comments. By addressing them, we believe we've substantially strengthened support for our conclusions.

Major issues:

1) Functional characterization of candidate drivers and mutations
The authors claim that "Driver mutations identify oncogenic lncRNAs with therapeutic potential" (l. 245), which is not currently justified in this generality by the data presented:

1a) The authors analyze in total three lncRNAs (MILC, MIHNC, NEAT1) out of the 54 mentioned in the abstract - which is a very low number and should be expanded especially given the ease to perturb these transcripts in tumor cell lines and determine cellular phenotypes like proliferation.

To gain insights into the general relevance of our findings, we have extended the functional validation to 9 candidate driver lncRNAs. Our new results show that 3/9 of these significantly enhance cell growth when overexpressed in an unmatched HeLa background (Figure R3.1 below).

Interestingly, overexpression of three driver candidates yielded a growth phenotype in a cell line matched to its tumour of origin (MIHNC1 & 2 in Head and neck, MILC in Liver), but MIHNC1 showed no effect in HeLa. This suggests that our HeLa results are likely an underestimate of the true number of driver lncRNAs that can promote cell proliferation, and overexpression in matched cell backgrounds would likely identify more fitness altering lncRNAs.

These findings support the idea that lncRNAs discovered by driver analysis contain useful, novel cancer relevant genes.

We have described these new data in the Results and Figures (Figure 4a and Supplementary Figure S4a):

"Thus, we overexpressed a panel of nine candidates in HeLa cells and found that three promote cell growth (Figure 4a, Supplementary Figure S4a). It was interesting to note that, amongst the six that did not display a significant effect was a lncRNA, AC087463.1, herein

named MIHNC1 (Mutated in Head and Neck Cancer), which appeared as a potential driver in the Head and Neck (HN) tumour cohort (Figure 3b)."

a Viability assay, HeLa

Figure R3.1. Functional validation of driver lncRNA candidates. All experiments were performed by overexpressing indicated lncRNAs by plasmid vectors in HeLa cells. Upper panel: Candidates that significantly increased cell proliferation. Lower panel: Candidates that did not significantly increase cell proliferation.

1b) In fact, one would understand and expect from the abstract that all 54 lncRNAs ("...display oncogenic potential when experimentally expressed...") would have been tested for their oncogenic potential - which is certainly not necessary, but should be clarified in the abstract.

We agree and have clarified the language in the Abstract to avoid giving a false impression that all 54 lncRNAs have been validated.

1c) The differences in the proliferation potential are moderate at best for MILC (Figures 4d, S4d) - transformation assays in non-transformed cells that would be closer to the determination of oncogenicity have not been performed. Hence, their functional relevance can be claimed, but it remains unanswered whether these are oncogenic drivers.

This is an excellent suggestion, which we have addressed using two distinct models: immortalised human hepatocytes (IHH) and primary human hepatocytes.

Firstly, in the non-transformed cell-line background represented by IHH, we observed overexpression of MILC leads to increased cell viability in a non-transformed hepatocyte background (Figure R3.2). This information has been updated in the main text and figures (Figure 4l,m).

Figure R3.2: Effect of overexpression of mutant MILC lncRNA in immortalised human hepatocytes (IHH). Left panel: Expression of MILC as estimated by qRT-PCR. Right panel: Cell viability. Control: empty expression plasmid; Mutant: expression plasmid containing MILC with observed tumour mutations as defined in Main Figure 4d.

Figure R3.3: Effect of overexpression of mutant MILC lncRNA in primary hepatocytes from a human donor. Left: Transduction efficiency was evaluated using the plasmid-encoded GFP marker; Right: qRT-PCR was used to measure proliferation-associated cytokine mRNAs in untreated cells, and cells transduced with an empty lentivirus or one expressing mutated MILC.

1d) For the overexpression experiment, the authors combined four (MILC) or five different (MINHC) mutations in one transcript rather than testing them individually. Have these four or five mutations been found together in one allele of one patient? Otherwise, these would need to be tested individually to at least get one step closer to the situation in the patient and see whether individual mutations indeed have an impact.

This is an important point, regarding the effect of mutations from individual patients. The experiments in the manuscript were performed with mutations aggregated from several patients. To address the question of whether mutations from an individual patient can yield an observable effect, we have performed additional experiments. We designed new expression constructs that only contain mutations from individual patients, and repeated the same experiments with these constructs. We summarise the results here:

MILC: We confirmed our previous observation, that four mutations together from >1 patient led to increased oncogenic activity. However, for mutations from individual patients (Figure R3.4), while we observed similar trends, none of the individual patient mutations (i.e. Mut1, including two mutations identified within the same patient, indicated in purple boxes in Figure R3.4 and Mut2, comprising three mutations from the same patient, grey boxes) reached statistical significance (Figure R3.5). We interpret this to mean that individual mutations are relatively weak in effect, consistent with the Weak Driver hypothesis (Kumar et al, 2020), and that our in vitro assays may be insufficiently sensitive to distinguish those effects. It should be noted that tumour mutations have many years to exert their fitness effects within a patient, while our experiments take place over timeframes of days. We also reiterate that the observed mutations are highly recurrent between patients and tumour types (Figure R3.4). We believe that these findings are interesting and worth commenting on in the paper. These new data have been included in updated Supplementary Figure S4f and the main text.

“Having established that MILC promotes cell growth in a number of backgrounds, we next asked whether tumour mutations can enhance this activity, as would be expected for driver mutations. To do so, we designed and validated overexpression plasmids for the wild-type or mutated forms of the transcript (Figure 4j). We first tested mutations from two individual patients separately. We selected mutations that were recurrently observed in independent tumours from both PCAWG and HFM dataset (Figure 4d) (i.e. Mut1, including two mutations identified within the same patient and Mut2, including three mutations from the same patient, grey boxes, depicted in Supplementary Figure S4g). Importantly, transfection of wild-type MILC boosted cell growth, consistent with ASO results above (Supplementary Figure S4f, WT). However, neither of the individual patients’ mutations alone yielded statistically-significant changes in cell growth (Supplementary Figure S4f).”

Figure R3.4: Mutations in MILC lncRNA. Boxed mutations were tested in the indicated constructs. Mut1 mutations all derived from the same patient, while Mut2 mutation from another patient.

Figure R3.5: Cell viability of HuH7 cells 120h after transfection with plasmids expressing indicated variants of MILC lncRNA. Shown are the mean and standard deviation of $n \geq 6$ replicates. See Figure R3.4 for guide to mutations in overexpression constructs.

MIHNC1: We performed similar experiments to overexpress variants of MIHNC1 carrying mutations from individual patients. Unfortunately, we observed a high degree of variability in effects due to these mutations. Therefore, after careful consideration, we have decided to remove these experiments from the manuscript and plan to perform follow up work to better characterise the effect of somatic mutations in this gene. Nevertheless, it is important to clarify that the pro-fitness effect of wild-type MIHNC1 is not in question, therefore we are confident to include these data in the updated Figure 4b,c.

1e) For a "therapeutic potential", much more data would be needed including e.g. a therapeutic window between non-transformed and tumor cells, which is not addressed in the study - and also not necessary. However, all claims regarding therapeutic potential should be removed from the manuscript.

We agree and have removed all reference to therapeutic potential from the manuscript.

2) NEAT1
The mutations in the gene for the lncRNA NEAT1 are characterized in greater detail in a model system harboring mutations endogenously introduced by the CRISPR/Cas9 approach.

2a) Mutations have been introduced by non-homologous end joining repair and thus randomly at the selected targeted sites rather than by homologous recombination or prime editing resulting in precision genome editing. This raises the question how well these mixtures of mutations mimic the mutations found in patients. Are these specific mutations also found to a similar extent in patients? Preferentially, precision genome editing for at least the most frequent mutations in NEAT1 would be desirable to analyze actual patient-derived mutations in clonal cells.

Thank you for this suggestion of inserting precise mutations to replicate observed SNVs. While we were careful to validate that our CRISPR method yielded naturalistic small mutations at target sites (Figure 5c), nonetheless it would be ideal to test precise mutations in cellular DNA. To address this, we have invested considerable effort in using CRISPR homologous recombination (HR) and CRISPR Prime Editing (PE). We briefly summarise

the results here:

HR: We performed these experiments using 4 SNVs from NEAT1 in HeLa and HCT116 cells, and we used NGS targeted sequencing to evaluate the rate of successful precise editing. Unfortunately, the rate of insertion was extremely low (0.26% for Reg2 SNV of the paper; column “HDR” from Figure R3.6), and we observed a high degree of fluctuation in SNV detection at subsequent timepoints. Until the efficiency is increased, we don’t believe its possible to make quantitative estimates of fitness effects using HDR.

Figure R3.6: Next generation sequencing was used to evaluate the efficiency of editing by homologous recombination. y-axis: rate of editing, %. x-axis: observed edits. The desired edits are found in the column “HDR” (arrow).

PE: We also attempted to introduce NEAT1 SNVs using CRISPR Prime Editing. Altogether we tested 7 pegRNAs targeting 2 SNVs from NEAT1 in HCT116 and HeLa cells. We applied two different available PE enzymes, PE2 and PEMax. In preliminary experiments, we successfully introduced positive control mutations in the HEK3 gene locus. However, in the case of NEAT1 mutations, Sanger sequencing showed no successful editing outcomes (Figure R3.7).

Figure R3.7: HCT116 cells were transfected with PE2 or PEMax Cas9 systems and expression validated by fluorescent markers. Sanger chromatograms show no successful editing peaks when targeting NEAT1.

We intend to deploy CRISPR methods to introduce SNVs and evaluate their fitness effects in the future. However within the timeframe of this publication it will not be feasible to execute these complex techniques with the required level of care and attention to detail.

2b) The CRISPR mutagenesis approach may also lead to off-target effects which have not been controlled for. Although multiple different sgRNAs have been used, a rescue experiment (e.g. knockdown) would thus be desirable for at least one selected mutation.

a) KO cell lines

Thank you for suggesting this important control. To address it, we turned to NEAT1 knockout (KO) cells, where perturbation of NEAT1 should display no phenotypic effect. We treated NEAT1 KO cells with control sgRNA (Control2) or NEAT1 mutations in Region 2 (Reg2, sgRNA1) and Region 3 (Reg3, sgRNA7) and we measured the cell viability after 96 and 120 h. No differences were observed between cells treated with sgRNAs targeting NEAT1 and control sgRNA (Figure R3.8). The data have been included in Supplementary Figure 5d and described in the main text. These data are consistent with observed cellular phenotypes occurring via NEAT1 mutations, and not through off-target effects.

Figure R3.8: Viability of NEAT1 knockout (KO) HeLa cells, in response to treatment with Control2 and two indicated NEAT1-targeting sgRNAs. Cell viability (y-axis) was measured using Cell Titre Glo assay at two timepoints following sgRNA delivery (96 / 120 hours).

2c) In the interaction analysis, a fragment of only 288 nt is found to specifically bind to 154 different proteins. Then, changing only two nucleotides leads to changes for 26 proteins. Additional experiments should corroborate these findings. For example, a RIP experiment could be performed for two interactors - one enriched and one depleted - with wild-type and mutant NEAT1 transcripts to verify their differential interaction.

Thank you for suggesting this important control.

We have performed the suggested experiments. Using antibodies for one mutant-depleted protein (SREK1) and one mutant-enriched (PQBP1), we performed replicated RNA immunoprecipitation (RIP) experiments with wild-type (WT) and mutant (MUT) NEAT1 RNA (Figure R3.9). Firstly, we observe binding of NEAT1 sequence to both proteins over

background (IgG), as expected. Secondly, we observe changes in binding that correspond exactly with the mass spectrometry experiment: SREK binds more to WT NEAT1, and PQBP1 binds more to MUT NEAT1. These data have been included as a new Figure 6h in the manuscript.

Figure R3.9: RNA immunoprecipitation of NEAT1-bound proteins. Cells were transfected with vectors expressing wild-type (WT) or mutated (MUT) NEAT1 sequences. Immunoprecipitations were performed with antibodies against the indicated proteins. Control IgG antibody was used to estimate background.

2d) In figure 5h, significance levels should be indicated. The relevance of figure 6f remains unclear.

Figure 5h: Significance values have now been included.

Figure 6f: We have replaced this panel with a new cartoon outlining our proposed mechanistic model for how NEAT1 mutations can impact cell phenotype via paraspeckles in the new Figure 6m.

Minor

issues:

3) Results
The manuscript and the practical use of the results would strongly benefit from a (supplementary) table presenting the most important results for the 54 lncRNAs selected including their name, identifier, localization, most important mutations, references on their function etc., so that all readers can immediately check for the lncRNAs of their interest.

We have now included thorough additional data for these lncRNAs with data from the LncCompare resource (Carlevaro-Fita et al, PMID31147707) in a new Supplementary File 6.

4) References
The references #39 and #49 seem to be incomplete or incorrectly formatted.
Done

5) Typos

Several figure labels harbor a number of typos which should be corrected (predicions, survivalal, etwork, fetures, recturm,...). Also the manuscript text should be checked again.

We have run a complete spell check now.

Reviewers' Comments:

Reviewer #2:

Remarks to the Author:

I appreciate the authors' thoughtful responses to my critique. I have no further concerns.

Reviewer #3:

Remarks to the Author:

The authors have fully addressed all my previous concerns and significantly strengthened their study by the addition of new data and a well balanced data interpretation making it now publishable.

Reviewer #4:

Remarks to the Author:

In this manuscript entitled "Tumour mutations in long noncoding RNAs enhance cell fitness", the authors performed a genome-wide analysis of fitness-altering SNVs across a cohort of 2583 primary and 3527 metastatic tumours that have whole genome data available. The analysis identified mutated lncRNAs that are significantly enriched for previously-reported cancer genes. They then used CRISPRa technology to demonstrate that a number of these lncRNAs promote tumour cell proliferation. In addition to validate the overexpression of the identified lncRNAs can promoter tumour proliferation. They also experimentally validated a hotspot in the NEAT1 using in cellulo mutagenesis. They introduced the identified mutations in NEAT1 and observed a significant and reproducible increase in cell fitness, both in vitro and in a mouse model. Mechanistically, they have shown that NEAT1 SNVs remodel the interaction between NEAT1 and ribonucleoprotein and boost subnuclear paraspeckles. Overall, this study not only provided a software for lncRNA driver analysis to map cancer-promoting lncRNAs, but also experimentally validated that somatic mutations on lncRNAs can enhance the pathological cancer phenotypes. In the revised manuscript, the authors have added additional analysis, clarification, and experiments to address the reviewer #1's comments. The authors have released all the experiment details of the experiments to ensure the reproducibility and rigorous. I have no further concern.

Response to Reviewers' Comments

NCOMMS-22-11205A-Z

Tumour mutations in long noncoding RNAs enhance cell fitness

Esposito, Lanzós et al

Reviewer #2 (Remarks to the Author):

I appreciate the authors' thoughtful responses to my critique. I have no further concerns.

Reviewer #3 (Remarks to the Author):

The authors have fully addressed all my previous concerns and significantly strengthened their study by the addition of new data and a well balanced data interpretation making it now publishable.

Reviewer #4, computational analysis of long noncoding RNAs to assess your responses to Reviewer's #1 previous comments (Remarks to the Author):

In this manuscript entitled "Tumour mutations in long noncoding RNAs enhance cell fitness", the authors performed a genome-wide analysis of fitness-altering SNVs across a cohort of 2583 primary and 3527 metastatic tumours that have whole genome data available. The analysis identified mutated lncRNAs that are significantly enriched for previously-reported cancer genes. They then used CRISPRa technology to demonstrate that a number of these lncRNAs promote tumour cell proliferation. In addition to validate the overexpression of the identified lncRNAs can promoter tumour proliferation. They also experimentally validated a hotspot in the NEAT1 using in cellulo mutagenesis. They introduced the identified mutations in NEAT1 and observed a significant and reproducible increase in cell fitness, both in vitro and in a mouse model. Mechanistically, they have shown that NEAT1 SNVs remodel the interaction between NEAT1 and ribonucleoprotein and boost subnuclear paraspeckles. Overall, this study not only provided a software for lncRNA driver analysis to map cancer-promoting lncRNAs, but also experimentally validated that somatic mutations on lncRNAs can enhance the pathological cancer phenotypes. In the revised manuscript, the authors have added additional analysis, clarification, and experiments to address the reviewer #1's comments. The authors have released all the experiment details of the experiments to ensure the reproducibility and rigorous. I have no further concern.

Authors' comment:

We sincerely thank all the Reviewers for providing thoughtful feedbacks. Their useful suggestions allowed us thoroughly strengthened this work, further supporting our scientific conclusions.